# Improving Accelerated Federated Learning with Compression and Importance Sampling

## Abstract

Federated Learning is a collaborative training framework that leverages heterogeneous data distributed across a vast number of clients. Since it is practically infeasible to request and process all clients during the aggregation step, partial participation must be supported. In this setting, the communication between the server and clients poses a major bottleneck. To reduce communication loads, there are two main approaches: compression and local steps. Recent work by Mishchenko et al. (2022) introduced the new ProxSkip method, which achieves an accelerated rate using the local steps technique. Follow-up works successfully combined local steps acceleration with partial participation (Grudzień et al., 2023; Condat et al., 2023) and gradient compression (Condat et al., 2022). In this paper, we finally present a complete method for Federated Learning that incorporates all necessary ingredients: Local Training, Compression, and Partial Participation. We obtain state-of-the-art convergence guarantees in the considered setting. Moreover, we analyze the general sampling framework for partial participation and derive an importance sampling scheme, which leads to even better performance. We experimentally demonstrate the advantages of the proposed method in practice.

## 1 Introduction

Federated Learning (FL) (Konečný et al., 2016; McMahan & Ramage, 2017) is a distributed machine learning paradigm that allows multiple devices or clients to collaboratively train a shared model without transferring their raw data to a central server. In traditional machine learning, data is typically gathered and stored in a central location for training a model. However, in Federated Learning, each client trains a local model using its own data and shares only the updated model parameters with a central server or aggregator. The server then aggregates the updates from all clients to create a new global model, which is then sent back to each client to repeat the process (McMahan et al., 2016).

This approach has gained significant attention due to its ability to address the challenges of training machine learning models on decentralized and sensitive data McMahan et al. (2017). Federated Learning enables clients to preserve their privacy and security by keeping their data local and not sharing it with the central server. This approach also reduces the need for large-scale data transfers, thereby minimizing communication costs and latency (Li et al., 2020a).

Federated Learning poses several challenges such as data heterogeneity, communication constraints, and ensuring the privacy and security of the data (Kairouz et al., 2021). Researchers in this field have developed novel optimization algorithms to address these challenges and to enable efficient aggregation of the model updates from multiple clients (Wang et al., 2021b). Federated Learning has been successfully applied to various applications, including healthcare (Vepakomma et al., 2018), finance (Long et al., 2020), and Internet of Things (IoT) devices (Khan et al., 2021).

We consider the standard formulation of Federated Learning as a finite sum minimization problem:

$$\min_{x \in \mathbb{R}^d} \left[ f(x) \coloneqq \frac{1}{M} \sum_{m=1}^{M} f_m(x) \right] \tag{1}$$

where $M$ is the number of clients/devices. Each function $f_m(x) = \mathbb{E}_{\xi \sim \mathcal{D}_m}[l(x, \xi)]$ represents the average loss, calculated via the loss function $l$, of the model parameterized by $x \in \mathbb{R}^d$ over the training data $\mathcal{D}_m$ stored by client $m \in [M] \coloneqq \{1, \ldots, M\}$.

**Federated Averaging.** The method known as Federated Averaging (FedAvg), proposed by McMahan et al. (2017), addresses practical challenges in federated learning while solving problem 1. It builds upon Gradient Descent (GD) and incorporates four key modifications: Client Sampling (CS), Data Sampling (DS), Local Training (LT), and Communication Compression (CC).

The training process consists of multiple communication rounds. At each round $t$, a subset $S^t \subset [M]$ of clients, with size $C^t = |S^t|$, is chosen to participate. The server sends the current model $x^t$ to clients in $S^t$. Each client $m \in S^t$ performs $K$ iterations of Stochastic Gradient Descent (SGD) on their local loss function $f_m$, using mini-batches $\mathcal{B}_m^{k,t} \subseteq \mathcal{D}_m$ of size $b_m = |\mathcal{B}_m^{k,t}|$. Clients then compress and transmit their updates to the server for aggregation into a new model $x^{t+1}$, repeating the process. This scheme is described in Grudzień et al. (2023).

The four modifications can be independently activated or deactivated. For instance, when $C^t = M$ for all rounds, CS is deactivated. DS is disabled when $b_m = |D_m|$ for all clients, and LT is turned off when $K = 1$. If the compression operator is the identity, CC is deactivated. With all modifications disabled, FedAvg is equivalent to vanilla gradient descent (GD).

**Data Sampling.** Previous studies have highlighted the practical advantages of FedAvg but lack theoretical analysis. Given FedAvg's four distinct components, analyzing them separately is essential for a deeper understanding.

Unbiased data sampling techniques have a strong connection to stochastic approximation literature. For instance, CS mechanisms have been well-explored in both convex and nonconvex settings. Oracle-optimal versions of SGD supporting unbiased CS and DS mechanisms have been proposed, along with analyses using variance reduction techniques (Robbins & Monro, 1951; Nemirovsky & Yudin, 1983; Nemirovski et al., 2009; Bottou et al., 2018; Gower et al., 2019; Khaled & Richtárik, 2020; Tyurin et al., 2022a; Li et al., 2021; Fang et al., 2018; Nguyen et al., 2017a;b; Gorbunov et al., 2020).

**Client Sampling.** As distributed learning gained popularity, researchers investigated Client Sampling strategies for communication efficiency and security. Empirical and theoretical studies have examined optimal strategies under various conditions. While Client Sampling shares similarities with data sampling, it has distinct characteristics (Wu & Wang, 2022; So et al., 2021; Fraboni et al., 2021; Charles et al., 2021; Huang et al., 2022; Wang et al., 2022; Chen et al., 2022; Malinovsky et al., 2023; Cho et al., 2023).

**Communication Compression.** Compression techniques are crucial for distributed optimization, allowing clients to transmit compressed updates, reducing bandwidth usage. Various compression methods, including stochastic quantization and random sparsification, have been proposed. Unbiased compressors can reduce transmitted bits but may slow convergence due to increased gradient variance. To address this, Mishchenko et al. (2019) proposed DIANA, which uses control iterates to guarantee fast convergence. DIANA has been extended and applied in various scenarios. The article discusses the application of compression techniques in Federated Learning, including methods like compression with random reshuffling (Alistarh et al., 2017; Wangni et al., 2017; Stich et al., 2018; Tang et al., 2019; Khirirat et al., 2018; Stich, 2020; Mishchenko et al., 2019; Horváth et al., 2019; Safaryan et al., 2021; Wang et al., 2021a; Kovalev et al., 2021; Li et al., 2020c; Basu et al., 2019; Reisizadeh et al., 2020; Haddadpour et al., 2021; Khaled & Richtárik, 2019; Chraibi et al., 2019; Malinovsky & Richtárik, 2022; Sadiev et al., 2022b).

FIVE GENERATIONS OF LOCAL TRAINING

Local Training (LT) is a crucial aspect of Federated Learning (FL), where each participating client performs multiple local training steps before synchronization. In the smooth strongly convex regime, we provide an overview of advancements in understanding LT. (Malinovsky et al., 2022) categorized LT methods into five generations - heuristic, homogeneous, sublinear, linear, and accelerated.

**1st (Heuristic) Generation of LT Methods.** While the concepts of Local Training (LT) had been used in various machine learning fields (Povey et al., 2015; Moritz et al., 2016), it gained prominence as a communication acceleration technique with the introduction of the FedAvg algorithm (McMahan et al., 2017). However, this work, along with previous research, lacked theoretical justification. Hence, LT-based heuristics dominated the initial development of the field, devoid of theoretical guarantees.

**2nd (Homogeneous) Generation of LT Methods.** The second generation of LT methods provided guarantees but relied on data homogeneity assumptions. These assumptions included bounded gradients, requiring $\|\nabla f_m(x)\| \leq c$ for all $m \in [M]$ and $x \in \mathbb{R}^d$ (Li et al., 2020b), and bounded gradient dissimilarity, demanding $\frac{1}{M} \sum_{m=1}^{M} \|\nabla f_m(x)\|^2 \leq c\|\nabla f(x)\|^2$ for all $x \in \mathbb{R}^d$ (Haddadpour & Mahdavi, 2019). These assumptions aimed to exploit communication efficiency when local functions are identical. However, such assumptions are problematic and not met in many real-world cases. Relying on data/gradient homogeneity assumptions for analyzing LT methods is mathematically dubious and practically insignificant, given the non-i.i.d nature of Federated Learning datasets.

**3rd (Sublinear) Generation of LT Methods.** The third generation of LT methods eliminated the need for data homogeneity assumptions, as demonstrated by Khaled et al. (2019a;b). Nevertheless, studies by Woodworth et al. (2020b) and Glasgow et al. (2022) revealed that LocalGD with Data Sampling (LocalSGD) had communication complexity no better than minibatch SGD in heterogeneous data settings. Moreover, Malinovsky et al. (2020) analyzed LT methods for general fixed point problems, and Koloskova et al. (2020) studied decentralized aspects of Local Training. Despite removing data homogeneity assumptions, this generation showed pessimistic results, indicating sublinear rates for LocalGD, inferior to vanilla GD's linear convergence rate (Woodworth et al., 2020a).

**4th (Linear) Generation of LT Methods.** The fourth generation of LT methods aimed to develop linearly converging versions by addressing client drift issues identified in the previous generation. The Scaffold method, proposed by Karimireddy et al. (2020), successfully mitigated client drift and achieved a linear convergence rate. Other approaches, such as those by Gorbunov et al. (2021a) and Mitra et al. (2021), achieved similar results. While achieving linear convergence under standard assumptions was significant, these methods still had slightly higher communication complexity than vanilla GD and at best matched GD's complexity.

**5th (Accelerated) Generation of LT Methods.** Mishchenko et al. (2022) introduced the ProxSkip method, a new approach to Local Training that provably accelerates communication in the smooth strongly convex regime, even with heterogeneous data. Specifically, when each $f_m$ is $L$-smooth and $\mu$-strongly convex, ProxSkip can solve the optimization problem in $\mathcal{O}(\sqrt{L/\mu} \log 1/\varepsilon)$ communication rounds, a significant improvement over GD's $\mathcal{O}(L/\mu \log 1/\varepsilon)$ complexity. This accelerated communication complexity has been proven optimal (Scaman et al., 2019). Mishchenko et al. (2022) also introduced various extensions to ProxSkip, including flexible data sampling and a decentralized version. These developments led to the proposal of other methods for achieving communication acceleration through Local Training.

The initial article by Malinovsky et al. (2022) presents variance reduction for ProxSkip, while Condat & Richtárik (2022) applies ProxSkip to complex splitting schemes. Sadiev et al. (2022a) and Maranjyan et al. (2022) improve the computational complexity of ProxSkip. Condat et al. (2023) introduces accelerated Local Training methods with Client Sampling. Grudzień et al. (2023) provide an accelerated method with Client Sampling based on RandProx. CompressedScaffnew (Condat et al., 2022) achieves accelerated communication complexity using compression but requires permutation-based compressors (Szlendak et al., 2022).

## 2 CONTRIBUTIONS

Our work is based on the observation that none of the 5th generation Local Training (LT) methods currently support both Client Sampling (CS) and Communication Compression (CC). This raises the question of whether it is possible to design a method that can benefit from communication acceleration via LT while also supporting CS and utilizing Communication Compression techniques.

At this point, we are prepared to present the crucial observations and contributions made in our work.

- To the best of our knowledge, we provide the first LT method that successfully combines communication acceleration through local steps, Client Sampling techniques, and Communication Compression for a wide range of unbiased compressors. Our proposed algorithm for distributed optimization and federated learning is the first of its kind to utilize both strategies

Table 1: Comparison of local training (LT) methods.

| Method | Solver | DS | CS | VR [a] | CC | A-LT [b] | Sampling | Reference |
|---|---|---|---|---|---|---|---|---|
| Local-SGD | GD, SGD | ✓ | ✓ | ✗ | ✗ | ✗✗[c] | Uniform | Khaled et al. (2019b) |
| SCAFFOLD | GD, SGD | ✓ | ✓ | ✗ | ✗ | ✗ | Uniform | Karimireddy et al. (2020) |
| FedLin | GD, SGD | ✓ | ✗ | ✗ | ✓ | ✗ | ✗ | Mitra et al. (2021) |
| S-Local-SVRG | GD, SGD, VR-SGD | ✓ | ✗ | ✓ | ✗ | ✗ | ✗ | Gorbunov et al. (2021b) |
| ProxSkip | GD, SGD | ✓ | ✗ | ✗ | ✗ | ✓ | ✗ | Mishchenko et al. (2022) |
| ProxSkip-VR | GD, SGD, VR-SGD | ✓ | ✗ | ✓ | ✗ | ✓ | ✗ | Malinovsky et al. (2022) |
| APDA-Inexact | any | ✓ | ✗ | ✓ | ✗ | ✓ | ✗ | Sadiev et al. (2022a) |
| RandProx | Prox [d] | ✗ | ✓ | ✗ | ✓ | ✓ | Assumption 5 | Condat & Richtárik (2022) |
| TAMUNA | GD, SGD | ✓ | ✓ | ✗ | ✓[e] | ✓ | Uniform | Condat et al. (2023) |
| 5GCS | any | ✓ | ✓ | ✓ | ✗ | ✓ | Uniform | Grudzień et al. (2023) |
| 5GCS-AB | any | ✓ | ✓ | ✓ | ✗ | ✓ | Assumption (2) | This work |
| 5GCS-CC | any | ✓ | ✓ | ✓ | ✓[f] | ✓ | Uniform | This work |

[a] Supports variance-reduced DS on clients.
[b] Acceleration via local training.
[c] It has sublinear rate that is worse than GD rate.
[d] It requires exact calculations of proximal operators.
[e] TAMUNA supports only Perm-K Szlendak et al. (2022) compression.
[f] Any compressor satisfying Assumption (4).

in combination, resulting in a doubly accelerated rate. Our method based on method 5GCS (Grudzień et al., 2023) benefits from the two acceleration mechanisms provided by Local Training and compression in the Client Sampling regime, exhibiting improved dependency on the condition number of the functions and the dimension of the model, respectively.

- In this paper, we investigate a comprehensive Client Sampling framework based on the work of Tyurin et al. (2022b), which we then apply to the 5GCS method proposed by Grudzień et al. (2023). This approach enables us to analyze a wide range of Client Sampling techniques, including both sampling with and without replacement and it recovers previous results for uniform distribution. The framework also allows us to determine optimal probabilities, which results in improved communication.

## 3 PRELIMINARIES

**Method's description.** This section outlines the methods employed in this paper, focusing on two algorithms, Algorithm 1 and Algorithm 2, which share a common underlying concept. At the beginning of the training process, we initialize several parameters, including the starting point $x^0$, the dual (control) iterates $u_1^0, \ldots, u_M^0$, the primal (server-side) stepsize, and $M$ dual (local) stepsizes. Additionally, we choose a sampling scheme $\mathbf{S}$ for Algorithm 1 or a type of compressor $\mathcal{Q}$ for Algorithm 2. Once all parameters are set, we commence the iteration cycle.

At the start of each communication round, we sample a cohort (subset) of clients according to a particular scheme. The server then computes the intermediate model $\hat{x}^t$ and sends this point to each client in the cohort. Once each client receives the model $\hat{x}^t$, the worker uses it as a starting point for solving the local sub-problem defined in Equation 4. After approximately solving the local sub-problem, each client computes the gradient of the local function at the approximate solution $\nabla F_m(y_m^{K,t})$ and, based on this information, each client forms and sends an update to the server, with or without compression. The server aggregates the received information from workers and updates the global model $x^{t+1}$ and additional variables if necessary. This process repeats until convergence.

TECHNICAL ASSUMPTIONS

We begin by adopting the standard assumption in convex optimization (Nesterov, 2004).

**Assumption 1.** *The functions $f_m$ are $L_m$-smooth and $\mu_m$-strongly convex for all $m \in \{1, ..., M\}$.*

All of our theoretical results will rely on this standard assumption in convex optimization. To recap, a continuously differentiable function $\phi : \mathbb{R}^d \to \mathbb{R}$ is $L$-smooth if $\phi(x) - \phi(y) - \langle \nabla\phi(y), x - y \rangle \leq$

$\frac{L}{2}\|x-y\|^2$ for all $x, y \in \mathbb{R}^d$, and $\mu$-strongly convex if $\phi(x) - \phi(y) - \langle \nabla \phi(y), x - y \rangle \geq \frac{\mu}{2}\|x-y\|^2$ for all $x, y \in \mathbb{R}^d$, $\overline{L} = \frac{1}{M}\sum_{m=1}^M L_m$ and $L_{\max} = \max_m L_m$.

Our method employs the same reformulation of problem 1 as it is used in Grudzień et al. (2023), which we will now describe. Let $H : \mathbb{R}^d \to \mathbb{R}^{Md}$ be the linear operator that maps $x \in \mathbb{R}^d$ to the vector $(x, \ldots, x) \in \mathbb{R}^{Md}$ consisting of $M$ copies of $x$. First, note that $F_m(x) := \frac{1}{M}\left(f_m(x) - \frac{\mu_m}{2}\|x\|^2\right)$ is convex and $L_{F,m}$-smooth, where $L_{F,m} := \frac{1}{M}(L_m - \mu_m)$. Furthermore, we define $F : \mathbb{R}^{Md} \to \mathbb{R}$ as $F(x_1, \ldots, x_M) := \sum_{m=1}^M F_m(x_m)$.

Having introduced the necessary notation, we state the following formulation in the lifted space, which is equivalent to the initial problem 1:

$$x^\star = \underset{x \in \mathbb{R}^d}{\arg\min}\left[f(x) := F(Hx) + \frac{\mu}{2}\|x\|^2\right], \tag{2}$$

where $\mu = \frac{1}{M}\sum_{m=1}^M \mu_m$.

The dual problem to 2 has the following form:

$$u^\star = \underset{u \in \mathbb{R}^{Md}}{\arg\max}\left(\frac{1}{2\mu}\left\|\sum_{m=1}^M u_m\right\|^2 + \sum_{m=1}^M F_m^*(u_m)\right), \tag{3}$$

where $F_m^*$ is the Fenchel conjugate of $F_m$, defined by $F_m^*(y) := \sup_{x \in \mathbb{R}^d}\{\langle x, y \rangle - F_m(x)\}$. Under Assumption 1, the primal and dual problems have unique optimal solutions $x^\star$ and $u^\star$, respectively.

Next, we consider the tool of analyzing sampling schemes, which is Weighted AB Inequality from Tyurin et al. (2022b). Let $\Delta^M := \left\{(p_1, \ldots, p_M) \in \mathbb{R}^M \mid p_1, \ldots, p_M \geq 0, \sum_{m=1}^M p_m = 1\right\}$ be the standard simplex and $(\Omega, \mathcal{F}, \mathbf{P})$ a probability space.

**Assumption 2.** *(Weighted AB Inequality). Consider the random mapping* $\mathbf{S} : \{1, \ldots, M\} \times \Omega \to \{1, \ldots, M\}$*, which we call "sampling". For each sampling we consider the random mapping that we call estimator* $S : \mathbb{R}^d \times \ldots \times \mathbb{R}^d \times \Omega \to \mathbb{R}^d$*, such that* $\mathbb{E}[S(a_1, \ldots, a_M; \psi)] = \frac{1}{M}\sum_{m=1}^M a_m$ *for all* $a_1, \ldots, a_M \in \mathbb{R}^d$*. Assume that there exist* $A, B \geq 0$ *and weights* $(w_1, \ldots, w_M) \in \Delta^M$ *such that*

$$\mathbb{E}\left[\left\|S(a_1, \ldots, a_M; \psi) - \frac{1}{M}\sum_{m=1}^M a_m\right\|^2\right] \leq \frac{A}{M^2}\sum_{m=1}^M \frac{\|a_m\|^2}{w_m} - B\left\|\frac{1}{M}\sum_{m=1}^M a_m\right\|^2, \forall a_m \in \mathbb{R}^d.$$

Furthermore, it is necessary to specify the number of local steps to solve sub-problem 4. To maintain the generality and arbitrariness of local solvers, we use an inequality that ensures the accuracy of the approximate solutions of local sub-problems is sufficient. It should be noted that the assumption below covers a broad range of optimization methods, including all linearly convergent algorithms.

**Assumption 3.** *(Local Training). Let* $\{\mathcal{A}_1, \ldots, \mathcal{A}_M\}$ *be any Local Training (LT) subroutines for minimizing functions* $\{\psi_1^t, \ldots, \psi_M^t\}$ *defined in 4, capable of finding points* $\left\{y_1^{K,t}, \ldots, y_M^{K,t}\right\}$ *in* $K$ *steps, from the starting point* $y_m^{0,t} = \hat{x}^t$ *for all* $m \in \{1, \ldots, M\}$*, which satisfy the inequality*

$$\sum_{m=1}^M \frac{4}{\tau_m^2}\frac{\mu_m L_{F_m}^2}{3M}\left\|y_m^{K,t} - y_m^{\star,t}\right\|^2 + \sum_{m=1}^M \frac{L_{F_m}}{\tau_m^2}\left\|\nabla\psi_m^t(y_m^{K,t})\right\|^2 \leq \sum_{m=1}^M \frac{\mu_m}{6M}\left\|\hat{x}^t - y_m^{\star,t}\right\|^2,$$

*where* $y_m^{\star,t}$ *is the unique minimizer of* $\psi_m^t$*, and* $\tau_m \geq \frac{8\mu_m}{3M}$*.*

Finally, we need to specify the class of compression operators. We consider the class of unbiased compressors with conic variance (Condat & Richtárik, 2021).

**Assumption 4.** *(Unbiased compressor). A randomized mapping* $\mathcal{Q} : \mathbb{R}^d \to \mathbb{R}^d$ *is an unbiased compression operator* $(\mathcal{Q} \in \mathbb{U}(\omega)$ *for brevity) if for some* $\omega \geq 0$ *and* $\forall x \in \mathbb{R}^d$

$$\mathbb{E}\mathcal{Q}(x) = x, \text{ (Unbiasedness)} \quad \mathbb{E}\|\mathcal{Q}(x) - x\|^2 \leq \omega\|x\|^2 \text{ (Conic variance)}.$$

## 4 COMMUNICATION COMPRESSION

In this section we provide convergence guarantees for the Algorithm 1 (5GCS-CC), which is the version that combines Local Training, Client Sampling and Communication Compression.

**Algorithm 1** 5GCS-CC

1: **Input:** initial primal iterates $x^0 \in \mathbb{R}^d$; initial dual iterates $u_1^0, \ldots, u_M^0 \in \mathbb{R}^d$; primal stepsize $\gamma > 0$; dual stepsize $\tau > 0$; cohort size $C \in \{1, \ldots, M\}$
2: **Initialization:** $v^0 := \sum_{m=1}^M u_m^0$ $\qquad\qquad$ ⋄ The server initiates $v^0$ as the sum of the initial dual iterates
3: **for** communication round $t = 0, 1, \ldots$ **do**
4: $\quad$ Choose a cohort $S^t \subset \{1, \ldots, M\}$ of clients of cardinality $C$, uniformly at random $\qquad$ ⋄ CS step
5: $\quad$ Compute $\hat{x}^t = \frac{1}{1+\gamma\mu}\left(x^t - \gamma v^t\right)$ and broadcast it to the clients in the cohort
6: $\quad$ **for** $m \in S^t$ **do**
7: $\quad\quad$ Find $y_m^{K,t}$ as the final point after $K$ iterations of some local optimization algorithm $\mathcal{A}_m$, initiated with $y_m^0 = \hat{x}^t$, for solving the optimization problem $\qquad\qquad$ ⋄ Client $m$ performs $K$ LT steps

$$y_m^{K,t} \approx \arg\min_{y \in \mathbb{R}^d} \left\{ \psi_m^t(y) := F_m(y) + \frac{\tau_m}{2} \left\| y - \left( \hat{x}^t + \frac{1}{\tau_m} u_m^t \right) \right\|^2 \right\} \tag{4}$$

8: $\quad\quad$ Compute $\bar{u}_m^{t+1} = \nabla F_m(y_m^{K,t})$
9: $\quad\quad$ $u_m^{t+1} = u_m^t + \frac{1}{1+\omega}\frac{C}{M}Q_m\left(\bar{u}_m^{t+1} - u_m^t\right)$
10: $\quad\quad$ Send $Q_m\left(\bar{u}_m^{t+1} - u_m^t\right)$ to the server. $\qquad\qquad$ ⋄ Server updates $u_m^{t+1}$
11: $\quad$ **end for**
12:
13: $\quad$ **for** $m \in \{1, \ldots, M\}\backslash S^t$ **do**
14: $\quad\quad$ $u_m^{t+1} := u_m^t$ $\qquad\qquad\qquad\qquad\qquad$ ⋄ Non-participating clients do nothing
15: $\quad$ **end for**
16: $\quad$ $v^{t+1} := v^t + \frac{1}{1+\omega}\frac{C}{M}\sum_{m=1}^M \mathcal{Q}_m\left(\bar{u}_m^{t+1} - u_m^t\right)$ ⋄ The server keeps $v^{t+1}$ as the sum of the dual iterates
17: $\quad$ $x^{t+1} := \hat{x}^t - \gamma\frac{M}{C}(1+\omega)(v^{t+1} - v^t)$ $\qquad\qquad$ ⋄ The server updates the primal iterate
18: **end for**

**Theorem 4.1.** *Let Assumption 1 hold. Consider Algorithm 1 (5GCS-CC) with the LT solvers $\mathcal{A}_m$ satisfying Assumption 3 and compression operators $\mathcal{Q}_m$ satisfying Assumption 4. Let $\tau = \tau_m$ for all $m \in \{1, \ldots, M\}$ and $\frac{1}{\tau} - \gamma(M + \omega\frac{M}{C}) \geq \frac{4}{\tau^2}\frac{\mu}{3M}$, for example: $\tau \geq \frac{8\mu}{3M}$ and $\gamma = \frac{1}{2\tau(M+\omega\frac{M}{C})}$. Then for the Lyapunov function*

$$\Psi^t := \frac{1}{\gamma}\left\| x^t - x^\star \right\|^2 + \frac{M}{C}(\omega+1)\left(\frac{1}{\tau} + \frac{1}{L_{F,\max}}\right)\sum_{m=1}^M \left\| u_m^t - u_m^\star \right\|^2,$$

*the iterates satisfy $\mathbb{E}\left[\Psi^T\right] \leq (1-\rho)^T\Psi^0$, where $\rho := \min\left\{ \frac{\gamma\mu}{1+\gamma\mu}, \frac{C}{M(1+\omega)}\frac{\tau}{(L_{F,\max}+\tau)} \right\} < 1$.*

Next, we derive the communication complexity for Algorithm 1 (5GCS-CC).

**Corollary 4.2.** *Choose any $0 < \varepsilon < 1$ and $\tau = \frac{8}{3}\sqrt{\mu L_{\max}\left(\frac{\omega+1}{C}\right)\frac{1}{M(1+\frac{\omega}{C})}}$ and $\gamma = \frac{1}{2\tau M(1+\frac{\omega}{C})}$. In order to guarantee $\mathbb{E}\left[\Psi^T\right] \leq \varepsilon\Psi^0$, it suffices to take*

$$T \geq \mathcal{O}\left( \left(\frac{M}{C}(\omega+1) + \left(\sqrt{\frac{\omega}{C}} + 1\right)\sqrt{(\omega+1)\frac{M}{C}\frac{L}{\mu}}\right)\log\frac{1}{\varepsilon} \right)$$

*communication rounds.*

Note, if no compression is used ($\omega = 0$) we recover the rate of 5GCS: $\mathcal{O}\left( \left(M/C + \sqrt{ML/C\mu}\right)\log\frac{1}{\varepsilon} \right)$.

## 5 GENERAL CLIENT SAMPLING

In this section we analyze Algorithm 2 (5GCS-AB). First, we introduce a general result for all sampling schemes that can satisfy Assumption 2

**Theorem 5.1.** *Let Assumption 1 hold. Consider Algorithm 2 with sampling scheme $\mathbf{S}$ satisfying Assumption 2 and LT solvers $\mathcal{A}_m$ satisfying Assumption 3. Let the inequality hold $\frac{1}{\tau_m} - \left(\gamma(1-B)M + \gamma\frac{A}{w_m}\right) \geq \frac{4}{\tau_m^2}\frac{\mu_m}{3M}$, e.g. $\tau_m \geq \frac{8\mu_m}{3M}$ and $\gamma \leq \frac{1}{2\tau_m\left((1-B)M+\frac{A}{w_m}\right)}$. Then for the Lyapunov function*

$$\Psi^t := \frac{1}{\gamma}\left\| x^t - x^\star \right\|^2 + \sum_{m=1}^M (1+q_m)\left(\frac{1}{\tau_m} + \frac{1}{L_{F_m}}\right)\left\| u_m^t - u_m^\star \right\|^2,$$

---

**Algorithm 2** 5GCS-AB

---

1: **Input:** initial primal iterate $x^0 \in \mathbb{R}^d$; initial dual iterates $u_1^0, \ldots, u_M^0 \in \mathbb{R}^d$; primal stepsize $\gamma > 0$; dual stepsizes $\tau_m > 0$; $\omega \in \mathcal{D}_M$
2: **Initialization:** $v^0 := \sum_{m=1}^{M} u_m^0$      $\diamond$ The server initiates $v^0$ as the sum of the initial dual iterates
3: **for** communication round $t = 0, 1, \ldots$ **do**
4:   Sample a cohort $S^t \subset \{1, \ldots, M\}$ of clients according to sampling scheme **S**
5:   Compute $\hat{x}^t = \frac{1}{1+\gamma\mu} \left( x^t - \gamma v^t \right)$ and broadcast it to the clients in the cohort
6:   **for** $m \in S^t$ **do**
7:    Find $y_m^{K,t}$ as the final point after $K$ iterations of some local optimization algorithm $\mathcal{A}_m$, initiated with $y_m^0 = \hat{x}^t$, for solving the optimization problem     $\diamond$ Client $m$ performs $K$ LT steps

$$y_m^{K,t} \approx \arg\min_{y \in \mathbb{R}^d} \left\{ \psi_m^t(y) := F_m(y) + \frac{\tau_m}{2} \left\| y - \left( \hat{x}^t + \frac{1}{\tau_m} u_m^t \right) \right\|^2 \right\} \tag{6}$$

8:    Compute $\bar{u}_m^{t+1} = \nabla F_m(y_m^{K,t})$
9:    Update $u_m^{t+1} = \bar{u}_m^{t+1}$.
10:   **end for**
11:   **for** $m \in \{1, \ldots, M\} \setminus S^t$ **do**
12:    Update $u_m^{t+1} = u_m^t$.
13:   **end for**
14:   $x^{t+1} := \hat{x}^t - \gamma M \cdot S(u_1^{t+1} - u_1^t, \ldots, u_M^{t+1} - u_M^t; \omega)$    $\diamond$ The server updates the primal iterate
15:
16:   $v^{t+1} = \sum_{m=1}^{M} u_m^{t+1}$
17: **end for**

---

*the iterates of the method satisfy*

$$\mathbb{E}\left[ \Psi^{t+1} \right] \leq \max \left\{ \frac{1}{1+\gamma\mu}, \max_m \left[ \frac{L_{F_m} + \frac{q_m}{1+q_m}\tau_m}{L_{F_m} + \tau_m} \right] \right\} \mathbb{E}\left[ \Psi^t \right],$$

*where $q_m = \frac{1}{\widehat{p}_m} - 1$ and $\widehat{p}_m$ is probability that $m$-th client is participating.*

The obtained result is contingent upon the constants $A$ and $B$, as well as the weights $w_m$ specified in Assumption 2. Furthermore, the rate of the algorithm is influenced by $\widehat{p}_m$, which represents the probability of the $m$-th client participating. This probability is dependent on the chosen sampling scheme **S** and needs to be derived separately for each specific case. In main part of the work we consider two important examples: Multisampling and Independent Sampling.

## 5.1 SAMPLING WITH REPLACEMENT (MULTISAMPLING)

Let $\underline{p} = (p_1, p_2, \ldots, p_M)$ be probabilities summing up to 1 and let $\chi_m$ be the random variable equal to $m$ with probability $p_m$. Fix a cohort size $C \in \{1, 2, \ldots, M\}$ and let $\chi_1, \chi_2, \ldots, \chi_C$ be independent copies of $\chi$. Define the gradient estimator via

$$S\left( a_1, \ldots, a_n, \psi, \underline{p} \right) := \frac{1}{C} \sum_{m=1}^{C} \frac{a_{\chi_m}}{Mp_{\chi_m}}. \tag{5}$$

By utilizing this sampling scheme and its corresponding estimator, we gain the flexibility to assign arbitrary probabilities for client participation while also fixing the cohort size. However, it is important to note that under this sampling scheme, certain clients may appear multiple times within the cohort.

**Lemma 5.2.** *The Multisampling with estimator 5 satisfies the Assumption 2 with $A = B = \frac{1}{C}$ and $w_m = p_m$.*

Now we are ready to formulate the theorem.

**Theorem 5.3.** *Let Assumption 1 hold. Consider Algorithm 2 (5GCS-AB) with Multisampling and estimator 5 satisfying Assumption 2 and LT solvers $\mathcal{A}_m$ satisfying Assumption 3. Let the inequality hold $\frac{1}{\tau_m} - \left( \gamma \left( 1 - \frac{1}{C} \right) M + \gamma \frac{1}{Cp_m} \right) \geq \frac{4}{\tau_m^2} \frac{\mu_m}{3M}$ , e.g. $\tau_m \geq \frac{8\mu_m}{3M}$ and $\gamma \leq \frac{1}{2\tau_m\left( \left( 1 - \frac{1}{C} \right) M + \frac{1}{Cp_m} \right)}$. Then for the Lyapunov function*

$$\Psi^t := \frac{1}{\gamma} \left\| x^t - x^\star \right\|^2 + \sum_{m=1}^{M} \frac{1}{\widehat{p}_m} \left( \frac{1}{\tau_m} + \frac{1}{L_{F_m}} \right) \left\| u_m^t - u_m^\star \right\|^2,$$

*the iterates of the method satisfy*

$$\mathbb{E}\big[\Psi^{t+1}\big] \leq \max\left\{\frac{1}{1+\gamma\mu}, \max_m\left[\frac{L_{F_m}+(1-\widehat{p}_m)\tau_m}{L_{F_m}+\tau_m}\right]\right\}\mathbb{E}\big[\Psi^t\big],$$

*where* $\widehat{p}_m = 1-(1-p_m)^C$ *is probability that* $m$*-th client is participating.*

Regrettably, it does not appear to be feasible to obtain a closed-form solution for the optimal probabilities and stepsizes when $C > 1$. Nevertheless, we were able to identify a specific set of parameters for a special case where $C = 1$. Furthermore, even in this particular case, the solution is not exact. However, based on the Brouwer fixed-point theorem (Brouwer, 1911), a solution for $p_m$ and $\tau_m$ in Corollary 5.4 exists.

**Corollary 5.4.** *Suppose* $C = 1$. *Choose any* $0 < \varepsilon < 1$ *and* $p_m = \frac{\sqrt{L_{F,m}+\tau_m}}{\sum_{m=1}^M \sqrt{L_{F,m}+\tau_m}}$, *and* $\tau_m = \frac{8}{3}\sqrt{\overline{L}\mu M}p_m$. *In order to guarantee* $\mathbb{E}\big[\Psi^T\big] \leq \varepsilon\Psi^0$, *it suffices to take*

$$T \geq \max\left\{1 + \frac{16}{3}\sqrt{\frac{\overline{L}M}{\mu}}, \frac{3}{8}\sqrt{\frac{\overline{L}M}{\mu}} + M\right\}\log\frac{1}{\varepsilon}$$

*communication rounds.*

To address the challenge posed by the inexact solution, we have also included the exact formulas for the parameters. While this set of parameters may not offer the optimal complexity, it can still be valuable in certain cases.

**Corollary 5.5.** *Suppose* $C = 1$. *Choose any* $0 < \varepsilon < 1$ *and* $p_m = \frac{\sqrt{\frac{L_m}{M}}}{\sum_{m=1}^M \sqrt{\frac{L_m}{M}}}$, *and* $\tau_m = \frac{8}{3}\sqrt{\overline{L}\mu M}p_m$. *In order to guarantee* $\mathbb{E}\big[\Psi^T\big] \leq \varepsilon\Psi^0$, *it suffices to take*

$$T \geq \max\left\{1 + \frac{16}{3}\sqrt{\frac{\overline{L}M}{\mu}}, \frac{3}{8}\sqrt{\frac{\overline{L}M}{\mu}} + \frac{\sum_{m=1}^M \sqrt{L_m}}{\sqrt{L_{\min}}}\right\}\log\frac{1}{\varepsilon}$$

*communication rounds. Note that* $L_{\min} = \min_m L_m$.

## 5.2 SAMPLING WITHOUT REPLACEMENT (INDEPENDENT SAMPLING)

In the previous example, the server had the ability to control the cohort size and assign probabilities for client participation. However, in practical settings, the server lacks control over these probabilities due to various technical conditions such as internet connections, battery charge, workload, and others. Additionally, each client operates independently of the others. Considering these factors, we adopt the Independent Sampling approach. Let us formally define such a scheme. To do so, we introduce the concept of independent and identically distributed (i.i.d.) random variables:

$$\chi_m = \begin{cases} 1 & \text{with probability } p_m \\ 0 & \text{with probability } 1-p_m, \end{cases}$$

for all $m \in [M]$, also take $S^t := \{m \in [M]|\chi_m = 1\}$ and $\underline{p} = (p_1, \ldots, p_M)$. The corresponding estimator for this sampling has the following form:

$$S(a_1, \ldots, a_M, \psi, \underline{p}) := \frac{1}{M}\sum_{m \in S}\frac{a_m}{p_m}, \tag{7}$$

The described sampling scheme with its estimator is called the Independence Sampling. Specifically, it is essential to consider the probability that all clients communicate, denoted as $\Pi_{m=1}^M p_m$, as well as the probability that no client participates, denoted as $\Pi_{m=1}^M(1-p_m)$. It is important to note that $\sum_{m=1}^M p_m$ is not necessarily equal to 1 in general. Furthermore, the cohort size is not fixed but rather random, with the expected cohort size denoted as $\mathbb{E}[S^t] = \sum_{m=1}^M p_m$.

**Lemma 5.6.** *The Independent Sampling with estimator 7 satisfies the Assumption 2 with* $A = \frac{1}{\sum_m^M \frac{p_m}{1-p_m}}$, $B = 0$ *and* $w_m = \frac{\frac{p_m}{1-p_m}}{\sum_{m=1}^M \frac{p_m}{1-p_m}}$.

Now we are ready to formulate the convergence guarantees and derive communication complexity.

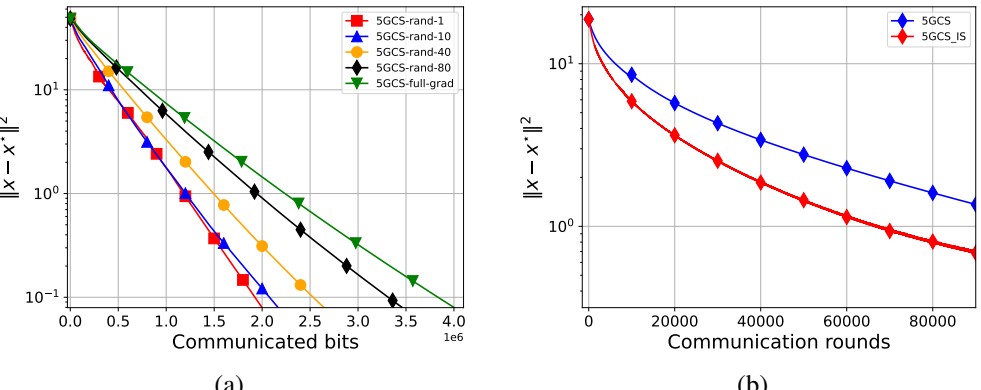

(a)                                        (b)

Figure 1: (a) Performance of Algorithm 1 (5GCS-CC) with different levels of sparsification $k$. (b) Comparison of Algorithm 2 (5GCS-AB) with uniform sampling and Multisampling in case of $C = 1$.

**Corollary 5.7.** *Choose any $0 < \varepsilon < 1$ and $p_m$ can be estimated but not set, then set $\tau_m = \frac{8}{3}\sqrt{\frac{\bar{L}\mu}{M\sum_{m=1}^{M}p_m}}$ and $\gamma = \frac{1}{2\tau_m\left(M+\frac{1-p_m}{p_m}\right)}$. In order to guarantee $\mathbb{E}\left[\Psi^T\right] \leq \varepsilon\Psi^0$, it suffices to take*

$$T \geq \max\left\{1 + \frac{16}{3}\sqrt{\frac{\overline{L}M}{\mu\sum_{m=1}^{M}p_m}}\left(1 + \frac{1-p_m}{Mp_m}\right), \max_m\left[\frac{3L_{F_m}}{8p_m}\sqrt{\frac{M\sum_{m=1}^{M}p_m}{\overline{L}\mu}} + \frac{1}{p_m}\right]\right\}\log\frac{1}{\varepsilon}$$

*communication rounds.*

## 6 EXPERIMENTS

This study primarily focuses on analyzing the fundamental algorithmic and theoretical aspects of a particular class of algorithms, rather than conducting extensive large-scale experiments. While we acknowledge the importance of such experiments, they fall outside the scope of this work. Instead, we provide illustrative examples and validate our findings through the application of logistic regression to a practical problem setting.

We are considering $\ell_2$-regularized logistic regression, which is a mathematical model used for classification tasks. The objective function, denoted as $f(x)$, is defined as follows:

$$f(x) = \frac{1}{MN}\sum_{m=1}^{M}\sum_{i=1}^{N}\log\left(1 + e^{-b_{m,i}a_{m,i}^{\top}x}\right) + \frac{\lambda}{2}\|x\|^2.$$

In this equation, $a_{m,i} \in \mathbb{R}^d$ and $b_{m,i} \in \{-1, +1\}$ represent the data samples and labels, respectively. The variables $M$ and $N$ correspond to the number of clients and the number of data points per client, respectively. The term $\lambda$ is a regularization parameter, and in accordance with Condat et al. (2023), we set $\lambda$, such that we have $\kappa = 10^4$. To illustrate our experimental results, we have chosen to focus on a specific case using the "a1a" dataset from the LibSVM library (Chang & Lin, 2011). We have $d = 119$, $M = 107$ and $N = 15$ for this dataset.

For the experiments involving communication compression, we utilized the Rand-$k$ compressor (Mishchenko et al., 2019) with various parameters for sparsification and theoretical stepsizes for the method. Based on the plotted results, it is evident that the optimal choice is achieved when setting $k = 1$ and the method without communication compression shows the worst performance. We calculate the number of communicated floats by all clients. In the experiments conducted to evaluate the Multisampling strategy, we employed the exact version of the parameters outlined in Corollary 5.5. Additionally, we applied a re-scaling procedure to modify the distribution of $L_m$ in order to reduce its uniformity. The resulting values were approximately $L_{\min} \approx 1.48$ and $L_{\max} \approx 2 \cdot 10^4$.

The observed results indicate that the exact solution of determining probabilities and stepsizes., despite not being optimal, outperformed the version with uniform sampling.

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

# A APPENDIX

# B BASIC INEQUALITIES

## B.1 YOUNG'S INEQUALITIES

For all $x, y \in \mathbb{R}^d$ and all $a > 0$, we have

$$\langle x, y \rangle \leq \frac{a \|x\|^2}{2} + \frac{\|y\|^2}{2a}, \tag{8}$$

$$\|x + y\|^2 \leq 2 \|x\|^2 + 2 \|y\|^2, \tag{9}$$

$$\frac{1}{2} \|x\|^2 - \|y\|^2 \leq \|x + y\|^2. \tag{10}$$

## B.2 VARIANCE DECOMPOSITION

For a random vector $X \in \mathbb{R}^d$ (with finite second moment) and any $c \in \mathbb{R}^d$, the variance of $X$ can be decomposed as

$$\mathbb{E}\left[ \|X - \mathbb{E}[X]\|^2 \right] = \mathbb{E}\left[ \|X - c\|^2 \right] - \|\mathbb{E}[X] - c\|^2. \tag{11}$$

## B.3 CONIC COMPRESSION VARIANCE

An unbiased randomized mapping $\mathcal{C} : \mathbb{R}^d \to \mathbb{R}^d$ has conic variance if there exists $\omega \geq 0$ such that

$$\mathbb{E}\left[ \|\mathcal{C}(x) - x\|^2 \right] \leq \omega \|x\|^2 \tag{12}$$

for all $x \in \mathbb{R}^d$.

## B.4 CONVEXITY AND $L$-SMOOTHNESS

Suppose $\phi \colon \mathbb{R}^d \to \mathbb{R}$ is $L$-smooth and convex. Then

$$\frac{1}{L} \|\nabla \phi(x) - \nabla \phi(y)\|^2 \leq \langle \nabla \phi(x) - \nabla \phi(y), x - y \rangle \tag{13}$$

for all $x, y \in \mathbb{R}^d$.

## B.5 DUAL PROBLEM AND SADDLE-POINT REFORMULATION

Then the saddle function reformulation of $(2)$ is:

$$\text{Find } (x^\star, (u_m^\star)_{m=1}^M) \in \arg \min_{x \in \mathbb{R}^d} \max_{u \in \mathbb{R}^{Md}} \left( \frac{\mu}{2} \|x\|^2 + \sum_{m=1}^M \langle x, u_m \rangle - \sum_{m=1}^M F_m^*(u_m) \right). \tag{14}$$

To ensure well-posedness of these problems, we need to assume that there exists $x^\star \in \mathbb{R}^d$ s.t.:

$$0 = \mu x^\star + \sum_{m=1}^M \nabla F_m(x^\star). \tag{15}$$

Which is equivalent to $(2)$, having a solution, which it does (unique in fact) as each $f_m$ is $\mu$-strongly convex. By first order optimality condition $x^\star$ and $u^\star$ that are solution to $(14)$, satisfy:

$$\begin{cases} 0 = \mu x^\star + \sum_{m=1}^M u_m^\star \\ H x^\star \in \partial F^*(u^\star) \end{cases}. \tag{16}$$

Where the latter in $(16)$ is equivalent to:

$$\nabla F(H x^\star) = u^\star. \tag{17}$$

Throughout, this section we will denote by $\mathcal{F}_t$ for all $t \geq 0$ the $\sigma$-algebra generated by the collection of $\left(\mathbb{R}^d \times \mathbb{R}^{dM}\right)$-valued random variables $\left(x^0, u^0\right), \ldots, \left(x^t, u^t\right)$.

## C  Proof of Theorem 5.1

**Theorem.** *Let Assumption 1 hold. Consider Algorithm 2 with sampling scheme $\mathbf{S}$ satisfying Assumption 2 and LT solvers $\mathcal{A}_m$ satisfying Assumption 3. Let the inequality hold $\frac{1}{\tau_m} - \left(\gamma\left(1 - B\right)M + \gamma\frac{A}{w_m}\right) \geq \frac{4}{\tau_m^2}\frac{\mu_m}{3M}$, e.g. $\tau_m \geq \frac{8\mu_m}{3M}$ and $\gamma \leq \frac{1}{2\tau_m\left((1-B)M+\frac{A}{w_m}\right)}$. Then for the Lyapunov function*

$$\Psi^t := \frac{1}{\gamma}\left\|x^t - x^\star\right\|^2 + \sum_{m=1}^{M}\left(1 + q_m\right)\left(\frac{1}{\tau_m} + \frac{1}{L_{F_m}}\right)\left\|u_m^t - u_m^\star\right\|^2,$$

*the iterates of the method satisfy*

$$\mathbb{E}\left[\Psi^{t+1}\right] \leq \max\left\{\frac{1}{1 + \gamma\mu}, \max_m\left[\frac{L_{F_m} + \frac{q_m}{1+q_m}\tau_m}{L_{F_m} + \tau_m}\right]\right\}\mathbb{E}\left[\Psi^t\right],$$

*where $q_m = \frac{1}{\widehat{p}_m} - 1$ and $\widehat{p}_m$ is probability that $m$-th client is participating.*

*Proof.* We start from using variance decomposition 11 and Proposition 1 from (Condat & Richtárik, 2021), we obtain

$$\mathbb{E}\left[\left\|x^{t+1} - x^\star\right\|^2 \mid \mathcal{F}_t\right] \overset{(11)}{=} \left\|\mathbb{E}\left[x^{t+1} \mid \mathcal{F}_t\right] - x^\star\right\|^2 + \mathbb{E}\left[\left\|x^{t+1} - \mathbb{E}\left[x^{t+1} \mid \mathcal{F}_t\right]\right\|^2 \mid \mathcal{F}_t\right]$$

$$\overset{(2)}{=} \underbrace{\left\|\hat{x}^t - x^\star - \gamma H^\top\left(\bar{u}^{t+1} - u^t\right)\right\|^2}_{X} - \gamma^2 B\left\|H^\top\left(\bar{u}^{t+1} - u^t\right)\right\|^2$$

$$+ \gamma^2\sum_{m=1}^{M}\frac{A}{w_m}\left\|\bar{u}_m^{t+1} - u_m^t\right\|^2. \tag{18}$$

Moreover, using (16) and the definition of $\hat{x}^t$, we have

$$(1 + \gamma\mu)\hat{x}^t = x^t - \gamma H^\top u^t, \tag{19}$$

$$(1 + \gamma\mu)x^\star = x^\star - \gamma H^\top u^\star. \tag{20}$$

Using (19) and (20) we obtain

$$
\begin{aligned}
X \quad = \quad & \left\|\hat{x}^t - x^\star\right\|^2 + \gamma^2\left\|H^\top\left(\bar{u}^{t+1} - u^t\right)\right\|^2 - 2\gamma\left\langle\hat{x}^t - x^\star, H^\top\left(\bar{u}^{t+1} - u^t\right)\right\rangle \\
\leq \quad & (1 + \gamma\mu)\left\|\hat{x}^t - x^\star\right\|^2 + \gamma^2\left\|H^\top\left(\bar{u}^{t+1} - u^t\right)\right\|^2 \\
& -2\gamma\left\langle\hat{x}^t - x^\star, H^\top\left(\bar{u}^{t+1} - u^\star\right)\right\rangle + 2\gamma\left\langle\hat{x}^t - x^\star, H^\top\left(u^t - u^\star\right)\right\rangle \\
\overset{(19)\pm(20)}{=} \quad & \left\langle x^t - x^\star - \gamma H^\top\left(u^t - u^\star\right), \hat{x}^t - x^\star\right\rangle + \gamma^2\left\|H^\top\left(\bar{u}^{t+1} - u^t\right)\right\|^2 \\
& -2\gamma\left\langle\hat{x}^t - x^\star, H^\top\left(\bar{u}^{t+1} - u^\star\right)\right\rangle + \left\langle\hat{x}^t - x^\star, 2\gamma H^\top\left(u^t - u^\star\right)\right\rangle \\
= \quad & \left\langle x^t - x^\star + \gamma H^\top\left(u^t - u^\star\right), \hat{x}^t - x^\star\right\rangle + \gamma^2\left\|H^\top\left(\bar{u}^{t+1} - u^t\right)\right\|^2 \\
& -2\gamma\left\langle\hat{x}^t - x^\star, H^\top\left(\bar{u}^{t+1} - u^\star\right)\right\rangle \\
\overset{(19)\pm(20)}{=} \quad & \frac{1}{1 + \gamma\mu}\left\langle x^t - x^\star + \gamma H^\top\left(u^t - u^\star\right), x^t - x^\star - \gamma H^\top\left(u^t - u^\star\right)\right\rangle \\
& +\gamma^2\left\|H^\top\left(\bar{u}^{t+1} - u^t\right)\right\|^2 - 2\gamma\left\langle\hat{x}^t - x^\star, H^\top\left(\bar{u}^{t+1} - u^\star\right)\right\rangle \\
= \quad & \frac{1}{1 + \gamma\mu}\left\|x^t - x^\star\right\|^2 - \frac{\gamma^2}{1 + \gamma\mu}\left\|H^\top\left(u^t - u^\star\right)\right\|^2 \\
& +\gamma^2\left\|H^\top\left(\bar{u}^{t+1} - u^t\right)\right\|^2 - 2\gamma\left\langle\hat{x}^t - x^\star, H^\top\left(\bar{u}^{t+1} - u^\star\right)\right\rangle. \tag{21}
\end{aligned}
$$

Combining (18) and (21)

$$\mathbb{E}\left[\left\|x^{t+1} - x^\star\right\|^2 \mid \mathcal{F}_t\right] \leq \frac{1}{1+\gamma\mu}\left\|x^t - x^\star\right\|^2 - \frac{\gamma^2}{1+\gamma\mu}\left\|H^\top(u^t - u^\star)\right\|^2$$
$$+\gamma^2(1-B)\left\|H^\top(\bar{u}^{t+1} - u^t)\right\|^2$$
$$-2\gamma\left\langle \hat{x}^t - x^\star, H^\top(\bar{u}^{t+1} - u^\star)\right\rangle$$
$$+\gamma^2 \sum_{m=1}^M \frac{A}{w_m}\left\|\bar{u}_m^{t+1} - u_m^t\right\|^2 - \frac{\gamma\mu}{M}\left\|H\hat{x}^t - Hx^\star\right\|^2.$$

Let $\widehat{p} = (\widehat{p}_1, \ldots, \widehat{p}_M)$. The update for $u$ may be written as

$$u_m^{t+1} = u_m^t + \widehat{p}_m \frac{1}{\widehat{p}_m} Bernoulli(\bar{u}_m^{t+1} - u_m^t, \widehat{p}_m),$$

where $\widehat{p}_m$ is the probability that client $m$ participates in the iteration. Firstly note that the update for $u_m^{t+1}$ can be written as:

$$u_m^{t+1} = u_m^t + \widehat{p}_m \widetilde{\mathcal{R}}_m(\bar{u}_m^{t+1} - u_m^t, \widehat{p}_m),$$

i.e we have a relation of $\frac{1}{1+q_m} = \widehat{p}_m$ , which obviously makes sense, since the independent, unbiased bernoulli compressor with probability $p_m$ has conic variance $q_m = \frac{1}{\widehat{p}_m} - 1$. This leads to

$$u_m^{t+1} = u_m^t + \frac{1}{1+q_m}\widetilde{\mathcal{R}}_m(\bar{u}_m^{t+1} - u_m^t, q_m).$$

Using such form, we get

$$\mathbb{E}\left[\left\|u_m^{t+1} - u_m^\star\right\|^2 \mid \mathcal{F}_t\right] \overset{(11)+(12)}{\leq} \left\|u_m^t - u_m^\star + \frac{1}{1+q_m}\left(\bar{u}_m^{t+1} - u_m^t\right)\right\|^2$$
$$+\frac{q_m}{(1+q_m)^2}\left\|\bar{u}_m^{t+1} - u_m^t\right\|^2$$
$$= \frac{q_m^2}{(1+q_m)^2}\left\|u_m^t - u_m^\star\right\|^2 + \frac{1}{(1+q_m)^2}\left\|\bar{u}_m^{t+1} - u_m^\star\right\|^2$$
$$+\frac{2q_m}{(1+q_m)^2}\left\langle u_m^t - u_m^\star, \bar{u}_m^{t+1} - u_m^\star\right\rangle$$
$$+\frac{q_m}{(1+q_m)^2}\left\|\bar{u}_m^{t+1} - u_m^\star\right\|^2 + \frac{q_m}{(1+q_m)^2}\left\|u_m^t - u_m^\star\right\|^2$$
$$-\frac{2q_m}{(1+q_m)^2}\left\langle u_m^t - u_m^\star, \bar{u}_m^{t+1} - u_m^\star\right\rangle$$
$$\leq \frac{1}{1+q_m}\left\|\bar{u}_m^{t+1} - u_m^\star\right\|^2 + \frac{q_m}{1+q_m}\left\|u_m^t - u_m^\star\right\|^2. \quad (22)$$

Let us consider the first term in (22):

$$\left\|\bar{u}_m^{t+1} - u_m^\star\right\|^2 = \left\|(u_m^t - u_m^\star) + (\bar{u}_m^{t+1} - u_m^t)\right\|^2$$
$$= \left\|u_m^t - u_m^\star\right\|^2 + \left\|\bar{u}_m^{t+1} - u_m^t\right\|^2 + 2\left\langle u_m^t - u_m^\star, \bar{u}_m^{t+1} - u_m^t\right\rangle$$
$$= \left\|u_m^t - u_m^\star\right\|^2 + 2\left\langle \bar{u}_m^{t+1} - u_m^\star, \bar{u}_m^{t+1} - u_m^t\right\rangle - \left\|\bar{u}_m^{t+1} - u_m^t\right\|^2.$$

Combining terms together we get

$$\mathbb{E}\left[\left\|u_m^{t+1} - u_m^\star\right\|^2 \mid \mathcal{F}_t\right] \leq \left\|u_m^t - u_m^\star\right\|^2$$
$$+\frac{1}{1+q_m}\left(2\left\langle \bar{u}_m^{t+1} - u_m^\star, \bar{u}_m^{t+1} - u_m^t\right\rangle - \left\|\bar{u}_m^{t+1} - u_m^t\right\|^2\right).$$

Finally, we obtain

$$
\begin{aligned}
\frac{1}{\gamma}\mathbb{E}\Big[\big\|x^{t+1} - x^\star\big\|^2 \mid \mathcal{F}_t\Big] \;+\;& \sum_{m=1}^{M} \frac{1+q_m}{\tau_m}\mathbb{E}\Big[\big\|u_m^{t+1} - u_m^\star\big\|^2 \mid \mathcal{F}_t\Big] \\
\leq\;& \frac{1}{\gamma\left(1+\gamma\mu\right)}\big\|x^t - x^\star\big\|^2 - \frac{\gamma}{1+\gamma\mu}\big\|H^\top(u^t - u^\star)\big\|^2 \\
&+\gamma(1-B)\big\|H^\top(\bar{u}^{t+1} - u^t)\big\|^2 \\
&+\gamma\sum_{m=1}^{M}\frac{A}{w_m}\big\|\bar{u}_m^{t+1} - u_m^t\big\|^2 - \frac{\mu}{M}\big\|H\hat{x}^t - Hx^\star\big\|^2 \\
&+\frac{1+q_m}{\tau_m}\big\|u_m^t - u_m^\star\big\|^2 - 2\sum_{m=1}^{M}\big\langle \hat{x}^t - x^\star, \bar{u}_m^{t+1} - u_m^\star\big\rangle \\
&+\frac{1}{\tau_m}\left(2\big\langle \bar{u}_m^{t+1} - u_m^\star, \bar{u}_m^{t+1} - u_m^t\big\rangle - \big\|\bar{u}_m^{t+1} - u_m^t\big\|^2\right).
\end{aligned}
$$

Ignoring $-\frac{\gamma}{1+\gamma\mu}\big\|H^\top(u^t - u^\star)\big\|^2$ and noting

$$
\begin{aligned}
&-\big\langle \hat{x}^t - x^\star, \bar{u}_m^{t+1} - u_m^\star\big\rangle + \frac{1}{\tau_m}\big\langle \bar{u}_m^{t+1} - u_m^\star, \bar{u}_m^{t+1} - u_m^t\big\rangle \\
=\;& -\big\langle y_m^{K,t} - x^\star, \bar{u}_m^{t+1} - u_m^\star\big\rangle + \frac{1}{\tau_m}\big\langle \nabla\psi_m^t(y_m^{K,t}), \bar{u}_m^{t+1} - u_m^\star\big\rangle \\
\overset{(8)+(13)}{\leq}\;& -\frac{1}{L_{F_m}}\big\|\bar{u}_m^{t+1} - u_m^\star\big\|^2 + \frac{a_m}{2\tau_m}\big\|\nabla\psi_m^t(y_m^{K,t})\big\|^2 + \frac{1}{2a_m\tau_m}\big\|\bar{u}_m^{t+1} - u_m^\star\big\|^2 \\
=\;& -\left(\frac{1}{L_{F_m}} - \frac{1}{2a_m\tau_m}\right)\big\|\bar{u}_m^{t+1} - u_m^\star\big\|^2 + \frac{a_m}{2\tau_m}\big\|\nabla\psi_m^t(y_m^{K,t})\big\|^2 \\
\overset{(22)}{\leq}\;& -\left(\frac{1}{L_{F_m}} - \frac{1}{2a_m\tau_m}\right)\left((1+q_m)\mathbb{E}\Big[\big\|u_m^{t+1} - u_m^\star\big\|^2 \mid \mathcal{F}_t\Big] - q_m\big\|u_m^t - u_m^\star\big\|^2\right) \\
&+\frac{a_m}{2\tau_m}\big\|\nabla\psi_m^t(y_m^{K,t})\big\|^2,
\end{aligned}
$$

we get

$$
\begin{aligned}
\frac{1}{\gamma}\mathbb{E}\Big[\big\|x^{t+1} - x^\star\big\|^2 \mid \mathcal{F}_t\Big] \;+\;& \sum_{m=1}^{M}(1+q_m)\left(\frac{1}{\tau_m} + \frac{1}{L_{F_m}}\right)\mathbb{E}\Big[\big\|u_m^{t+1} - u_m^\star\big\|^2 \mid \mathcal{F}_t\Big] \\
\leq\;& \frac{1}{\gamma\left(1+\gamma\mu\right)}\big\|x^t - x^\star\big\|^2 \\
&+\sum_{m=1}^{M}(1+q_m)\left(\frac{1}{\tau_m} + \frac{q_m}{1+q_m}\frac{1}{L_{F_m}}\right)\big\|u_m^t - u_m^\star\big\|^2 \\
&+\sum_{m=1}^{M}\left(\gamma\left(1-B\right)M + \gamma\frac{A}{w_m} - \frac{1}{\tau_m}\right)\big\|\bar{u}_m^{t+1} - u_m^t\big\|^2 \\
&+\sum_{m=1}^{M}\frac{L_{F_m}}{\tau_m^2}\big\|\nabla\psi_m^t(y_m^{K,t})\big\|^2 - \sum_{m=1}^{M}\mu v_m\big\|\hat{x}^t - x^\star\big\|^2.
\end{aligned}
$$

Where we made the choice $a_m = \frac{L_{F_m}}{\tau_m}$ and $\sum_{m=1}^{M} v_m \leq 1$, positive real numbers, e.g. $\frac{\mu_m}{\sum_{m=1}^{M}\mu_m}$.
Using Young's inequality we have

$$
-\frac{\mu v_m}{3}\big\|\hat{x}^t - y_m^{\star,t} + y_m^{\star,t} - x^\star\big\|^2 \overset{(10)}{\leq} \frac{\mu v_m}{3}\big\|y_m^{\star,t} - x^\star\big\|^2 - \frac{\mu v_m}{6}\big\|\hat{x}^t - y_m^{\star,t}\big\|^2.
$$

Noting the fact that $y_m^{\star,t} = \hat{x}^t - \frac{1}{\tau_m}(\hat{u}_m^{t+1} - u_m^t)$, we have

$$
\frac{\mu v_m}{3}\big\|y_m^{\star,t} - x^\star\big\|^2 \overset{(9)}{\leq} 2\frac{\mu v_m}{3}\big\|\hat{x}^t - x^\star\big\|^2 + \frac{2}{\tau_m^2}\frac{\mu v_m}{3}\big\|\hat{u}_m^{t+1} - u_m^t\big\|^2.
$$

Combining those inequalities we get

$$
\frac{1}{\gamma}\mathbb{E}\Big[\big\|x^{t+1}-x^\star\big\|^2 \mid \mathcal{F}_t\Big] \;+\; \sum_{m=1}^{M}\left(1+q_m\right)\left(\frac{1}{\tau_m}+\frac{1}{L_{F_m}}\right)\mathbb{E}\Big[\big\|u_m^{t+1}-u_m^\star\big\|^2 \mid \mathcal{F}_t\Big]
$$

$$
\begin{aligned}
\leq\;& \frac{1}{\gamma\left(1+\gamma\mu\right)}\left\|x^t-x^\star\right\|^2 \\
&+\sum_{m=1}^{M}\left(1+q_m\right)\left(\frac{1}{\tau_m}+\frac{q_m}{1+q_m}\frac{1}{L_{F_m}}\right)\left\|u_m^t-u_m^\star\right\|^2 \\
&+\sum_{m=1}^{M}\frac{2}{\tau_m^2}\frac{\mu v_m}{3}\left\|\hat{u}_m^{t+1}-u_m^t\right\|^2 \\
&-\sum_{m=1}^{M}\left(\frac{1}{\tau_m}-\left(\gamma\left(1-B\right)M+\gamma\frac{A}{w_m}\right)\right)\left\|\bar{u}_m^{t+1}-u_m^t\right\|^2 \\
&+\sum_{m=1}^{M}\frac{L_{F_m}}{\tau_m^2}\left\|\nabla\psi_m^t(y_m^{K,t})\right\|^2-\sum_{m=1}^{M}\frac{\mu v_m}{6}\left\|\hat{x}^t-y_m^{\star,t}\right\|^2.
\end{aligned}
$$

Assuming $\gamma$ and $\tau_m$ can be chosen so that $\frac{1}{\tau_m}-\left(\gamma\left(1-B\right)M+\gamma\frac{A}{w_m}\right)\geq\frac{4}{\tau_m^2}\frac{\mu v_m}{3}$ we obtain

$$
\frac{1}{\gamma}\mathbb{E}\Big[\big\|x^{t+1}-x^\star\big\|^2 \mid \mathcal{F}_t\Big] \;+\; \sum_{m=1}^{M}\left(1+q_m\right)\left(\frac{1}{\tau_m}+\frac{1}{L_{F_m}}\right)\mathbb{E}\Big[\big\|u_m^{t+1}-u_m^\star\big\|^2 \mid \mathcal{F}_t\Big]
$$

$$
\begin{aligned}
\leq\;& \frac{1}{\gamma\left(1+\gamma\mu\right)}\left\|x^t-x^\star\right\|^2 \\
&+\sum_{m=1}^{M}\left(1+q_m\right)\left(\frac{1}{\tau_m}+\frac{q_m}{1+q_m}\frac{1}{L_{F_m}}\right)\left\|u_m^t-u_m^\star\right\|^2 \\
&+\sum_{m=1}^{M}\frac{4}{\tau_m^2}\frac{\mu v_m L_{F_m}^2}{3}\left\|y_m^{K,t}-y_m^{\star,t}\right\|^2 \\
&+\sum_{m=1}^{M}\frac{L_{F_m}}{\tau_m^2}\left\|\nabla\psi_m^t(y_m^{K,t})\right\|^2-\sum_{m=1}^{M}\frac{\mu v_m}{6}\left\|\hat{x}^t-y_m^{\star,t}\right\|^2.
\end{aligned}
$$

The point $y^{K,t}$ is supposed to satisfy Assumption 3:

$$
\sum_{m=1}^{M}\frac{4}{\tau_m^2}\frac{\mu v_m L_{F_m}^2}{3}\left\|y_m^{K,t}-y_m^{\star,t}\right\|^2+\sum_{m=1}^{M}\frac{L_{F_m}}{\tau_m^2}\left\|\nabla\psi_m^t(y_m^{K,t})\right\|^2\leq\sum_{m=1}^{M}\frac{\mu v_m}{6}\left\|\hat{x}^t-y_m^{\star,t}\right\|^2.
$$

Thus

$$
\frac{1}{\gamma}\mathbb{E}\Big[\big\|x^{t+1}-x^\star\big\|^2 \mid \mathcal{F}_t\Big] \;+\; \sum_{m=1}^{M}\left(1+q_m\right)\left(\frac{1}{\tau_m}+\frac{1}{L_{F_m}}\right)\mathbb{E}\Big[\big\|u_m^{t+1}-u_m^\star\big\|^2 \mid \mathcal{F}_t\Big]
$$

$$
\begin{aligned}
\leq\;& \frac{1}{\gamma\left(1+\gamma\mu\right)}\left\|x^t-x^\star\right\|^2 \\
&+\sum_{m=1}^{M}\left(1+q_m\right)\left(\frac{1}{\tau_m}+\frac{q_m}{1+q_m}\frac{1}{L_{F_m}}\right)\left\|u_m^t-u_m^\star\right\|^2.
\end{aligned}
$$

By taking the expectation on both sides we get

$$
\mathbb{E}\big[\Psi^{t+1}\big]\leq\max\left\{\frac{1}{1+\gamma\mu},\frac{L_{F_m}+\frac{q_m}{1+q_m}\tau_m}{L_{F_m}+\tau_m}\right\}\mathbb{E}\big[\Psi^t\big],
$$

which finishes the proof. $\qquad\square$

## D  MULTISAMPLING (SAMPLING WITH REPLACEMENT)

### D.1  PROOF OF LEMMA 5.2

**Lemma.** *The Multisampling with estimator 5 satisfies the Assumption 2 with $A = B = \frac{1}{C}$ and $w_m = p_m$.*

*Proof.* The proof is presented in Tyurin et al. (2022a). Let us provide it for completeness.

Let us fix $C > 0$. For all $m \in [C]$, we define i.i.d. random variables

$$
\mathcal{X}_m = \begin{cases} 1 & \text{with probability } p_1 \\ 2 & \text{with probability } p_2 \\ \quad . \\ \quad . \\ \quad . \\ M & \text{with probability } p_M, \end{cases}
$$

where $\underline{p} = (p_1, \ldots, p_M) \in \Delta^M$(simple simplex). A sampling

$$
S(a_1, \ldots, a_M; \underline{p}) \coloneqq \frac{1}{C} \sum_{m=1}^{C} \frac{a_{\mathcal{X}_m}}{M p_{\mathcal{X}_m}}
$$

is called the Importance sampling.

Let us establish inequality for Assumption 2:

$$
\mathbb{E}\left[\left\| \frac{1}{C} \sum_{m=1}^{C} \frac{a_{\mathcal{X}_m}}{M p_{\mathcal{X}_m}} - \frac{1}{M} \sum_{m=1}^{M} a_m \right\|^2\right] = \frac{1}{C^2} \sum_{m=1}^{C} \mathbb{E}\left[\left\| \frac{a_{\mathcal{X}_m}}{M p_{\mathcal{X}_m}} - \frac{1}{M} \sum_{m=1}^{M} a_m \right\|^2\right]
$$
$$
+ \frac{1}{C^2} \sum_{m \neq m'} \mathbb{E}\left[\left\langle \frac{a_{\mathcal{X}_m}}{M p_{\mathcal{X}_m}} - \frac{1}{M} \sum_{m=1}^{M} a_m, \frac{a_{\mathcal{X}'_m}}{M p_{\mathcal{X}'_m}} - \frac{1}{M} \sum_{m=1}^{M} a_m \right\rangle\right].
$$

By utilizing the independence and unbiasedness of the random variables, the final term becomes zero, resulting in:

$$
\begin{aligned}
\mathbb{E}\left[\left\| \frac{1}{C} \sum_{m=1}^{C} \frac{a_{\mathcal{X}_m}}{M p_{\mathcal{X}_m}} - \frac{1}{M} \sum_{m=1}^{M} a_m \right\|^2\right] &= \frac{1}{C^2} \sum_{m=1}^{C} \mathbb{E}\left[\left\| \frac{a_{\mathcal{X}_m}}{M p_{\mathcal{X}_m}} - \frac{1}{M} \sum_{m=1}^{M} a_m \right\|^2\right] \\
&= \frac{1}{C^2} \sum_{m=1}^{C} \mathbb{E}\left[\left\| \frac{a_{\mathcal{X}_m}}{M p_{\mathcal{X}_m}} \right\|^2\right] - \frac{1}{C} \left\| \frac{1}{M} \sum_{m=1}^{M} a_m \right\|^2 \\
&= \frac{1}{C} \sum_{m=1}^{M} p_m \left\| \frac{a_m}{M p_m} \right\|^2 - \frac{1}{C} \left\| \frac{1}{M} \sum_{m=1}^{M} a_m \right\|^2 \\
&= \frac{1}{C} \left( \frac{1}{M} \sum_{m=1}^{M} \frac{1}{M p_m} \|a_m\|^2 - \left\| \frac{1}{M} \sum_{m=1}^{M} a_m \right\|^2 \right).
\end{aligned}
$$

Thus we have $A = B = \frac{1}{C}$. $\qquad\qquad\square$

### D.2  PROOF OF THEOREM 5.3

**Theorem.** *Let Assumption 1 hold. Consider Algorithm 2 (5GCS-AB) with Multisampling and estimator 5 satisfying Assumption 2 and LT solvers $\mathcal{A}_m$ satisfying Assumption 3. Let the inequality*

*hold* $\frac{1}{\tau_m} - \left(\gamma\left(1 - \frac{1}{C}\right)M + \gamma\frac{1}{Cp_m}\right) \geq \frac{4}{\tau_m^2}\frac{\mu_m}{3M}$ , *e.g.* $\tau_m \geq \frac{8\mu_m}{3M}$ *and* $\gamma \leq \frac{1}{2\tau_m\left(\left(1-\frac{1}{C}\right)M + \frac{1}{Cp_m}\right)}$.
*Then for the Lyapunov function*

$$\Psi^t := \frac{1}{\gamma}\left\|x^t - x^\star\right\|^2 + \sum_{m=1}^{M}\frac{1}{\widehat{p}_m}\left(\frac{1}{\tau_m} + \frac{1}{L_{F_m}}\right)\left\|u_m^t - u_m^\star\right\|^2,$$

*the iterates of the method satisfy*

$$\mathbb{E}\left[\Psi^{t+1}\right] \leq \max\left\{\frac{1}{1+\gamma\mu}, \max_m\left[\frac{L_{F_m} + (1 - \widehat{p}_m)\tau_m}{L_{F_m} + \tau_m}\right]\right\}\mathbb{E}\left[\Psi^t\right],$$

*where* $\widehat{p}_m = 1 - (1 - p_m)^C$ *is probability that m-th client is participating.*

*Proof.* We start from theorem 5.1:

$$\mathbb{E}\left[\Psi^{t+1}\right] \leq \max\left\{\frac{1}{1+\gamma\mu}, \frac{L_{F_m} + \frac{q_m}{1+q_m}\tau_m}{L_{F_m} + \tau_m}\right\}\mathbb{E}\left[\Psi^t\right].$$

For Multisampling the probability of $m$-th client participating is $\widehat{p}_m = 1 - (1 - p_m)^C$ and we have relation $\widehat{p}_m = \frac{1}{1+q_m}$. Plugging $q_m = \frac{1}{\widehat{p}_m} - 1$ into recursion gives us

$$\mathbb{E}\left[\Psi^{t+1}\right] \leq \max\left\{\frac{1}{1+\gamma\mu}, \max_m\left[\frac{L_{F_m} + (1 - \widehat{p}_m)\tau_m}{L_{F_m} + \tau_m}\right]\right\}\mathbb{E}\left[\Psi^t\right].$$

Also using Lemma 5.2 we have $A = B = \frac{1}{C}$ and $w = p_m$. Plugging such constants to inequality for $\gamma$ and $\tau_m$ leads to $\frac{1}{\tau_m} - \left(\gamma\left(1 - \frac{1}{C}\right)M + \gamma\frac{1}{Cp_m}\right) \geq \frac{4}{\tau_m^2}\frac{\mu_m}{3M}$ , e.g. $\tau_m \geq \frac{8\mu_m}{3M}$ and $\gamma \leq \frac{1}{2\tau_m\left(\left(1-\frac{1}{C}\right)M + \frac{1}{Cp_m}\right)}$. $\qquad\square$

### D.3 PROOF OF COROLLARY 5.4

**Corollary.** *Suppose* $C = 1$. *Choose any* $0 < \varepsilon < 1$ *and* $p_m = \frac{\sqrt{L_{F,m}+\tau_m}}{\sum_{m=1}^{M}\sqrt{L_{F,m}+\tau_m}}$, *and* $\tau_m = \frac{8}{3}\sqrt{\overline{L}\mu M}p_m$. *In order to guarantee* $\mathbb{E}\left[\Psi^T\right] \leq \varepsilon\Psi^0$, *it suffices to take*

$$T \geq \max\left\{1 + \frac{16}{3}\sqrt{\frac{\overline{L}M}{\mu}}, \frac{3}{8}\sqrt{\frac{\overline{L}M}{\mu}} + M\right\}\log\frac{1}{\varepsilon}$$

*communication rounds.*

*Proof.* We set parameters as $p_m = \frac{\sqrt{L_{F_m}+\tau_m}}{\sum_{m=1}^{M}\sqrt{L_{F_m}+\tau_m}}$, and $\tau_m = \frac{8}{3}\sqrt{\overline{L}\mu M}p_m$. Let us derive the communication complexity:

$$T \geq \max\left\{1 + \frac{16}{3}\sqrt{\frac{\overline{L}M}{\mu}}, \frac{3}{8}\sqrt{\frac{\overline{L}M}{\mu}} + M\right\}\log\frac{1}{\varepsilon}$$

$$\geq \max\left\{1 + \frac{16}{3}\sqrt{\frac{\overline{L}M}{\mu}}, \frac{3}{8}\frac{M\overline{L} + M\frac{8}{3}\sqrt{\overline{L}\mu M}}{\sqrt{\overline{L}\mu M}}\right\}\log\frac{1}{\varepsilon}$$

$$\geq \max\left\{1 + \frac{16}{3}\sqrt{\frac{\overline{L}M}{\mu}}, \frac{3}{8}\frac{\left(\sum_{m=1}^{M}\sqrt{L_{F_m}+\tau_m}\right)^2}{\sqrt{\overline{L}\mu M}}\right\}\log\frac{1}{\varepsilon}$$

$$\geq \max\left\{1 + \frac{16}{3}\sqrt{\frac{\overline{L}M}{\mu}}, \max_m\left(\frac{3}{8}\frac{\left(\sum_{m=1}^{M}\sqrt{L_{F_m}+\tau_m}\right)^2}{(L_{F_m}+\tau_m)\sqrt{\overline{L}\mu M}}(L_{F_m}+\tau_m)\right)\right\}\log\frac{1}{\varepsilon}.$$

Unrolling the recursion from Theorem 5.3 we get

$$\mathbb{E}\left[\Psi^T\right] \leq \left(\max\left\{\frac{1}{1+\gamma\mu}, \max_m\left[\frac{L_{F_m} + (1-\widehat{p}_m)\,\tau_m}{L_{F_m} + \tau_m}\right]\right\}\right)^T \Psi^0. \tag{23}$$

Using Lemma from Malinovsky et al. (2021) for recursion (Appendix B), we can state that derived $T$ is sufficient to guarantee 23.

### D.4  PROOF OF COROLLARY 5.5

**Corollary.** *Suppose $C = 1$. Choose any $0 < \varepsilon < 1$ and $p_m = \frac{\sqrt{\frac{L_m}{M}}}{\sum_{m=1}^{M}\sqrt{\frac{L_m}{M}}}$, and $\tau_m = \frac{8}{3}\sqrt{\overline{L}\mu M}\,p_m$. In order to guarantee $\mathbb{E}\left[\Psi^T\right] \leq \varepsilon\Psi^0$, it suffices to take*

$$T \geq \max\left\{1 + \frac{16}{3}\sqrt{\frac{\overline{L}M}{\mu}}, \frac{3}{8}\sqrt{\frac{\overline{L}M}{\mu}} + \frac{\sum_{m=1}^{M}\sqrt{L_m}}{\sqrt{L_{\min}}}\right\}\log\frac{1}{\varepsilon}$$

*communication rounds. Note that $L_{\min} = \min_m L_m$.*

We set parameters as $p_m = \frac{\sqrt{\frac{L_m}{M}}}{\sum_{m=1}^{M}\sqrt{\frac{L_m}{M}}}$, and $\tau_m = \frac{8}{3}\sqrt{\overline{L}\mu M}\,p_m$. Let us derive the communication complexity. Since $\left(\sum_{m=1}^{M}\sqrt{\frac{L_m}{M}}\right)^2 \leq M\overline{L}$ we have

$$T \geq \max\left\{1 + \frac{16}{3}\sqrt{\frac{\overline{L}M}{\mu}}, \frac{3}{8}\sqrt{\frac{\overline{L}M}{\mu}} + \frac{\sum_{m=1}^{M}\sqrt{L_m}}{\sqrt{L_{\min}}}\right\}\log\frac{1}{\varepsilon}$$

$$\geq \max\left\{1 + \frac{16}{3}\sqrt{\frac{\overline{L}M}{\mu}}, \frac{3}{8}\frac{\left(\sum_{m=1}^{M}\sqrt{\frac{L_m}{M}}\right)^2}{\sqrt{\overline{L}\mu M}} + \frac{\sum_{m=1}^{M}\sqrt{\frac{L_m}{M}}}{\sqrt{\frac{L_{\min}}{M}}}\right\}\log\frac{1}{\varepsilon}$$

$$\geq \max\left\{1 + \frac{16}{3}\sqrt{\frac{\overline{L}M}{\mu}}, \max_m\left(\frac{3}{8}\frac{\left(\sum_{m=1}^{M}\sqrt{\frac{L_m}{M}}\right)^2}{\frac{L_m}{M}\sqrt{L^+\mu M}}\frac{L_m - \mu_m}{M} + \frac{\sum_{m=1}^{M}\sqrt{\frac{L_m}{M}}}{\sqrt{\frac{L_m}{M}}}\right)\right\}\log\frac{1}{\varepsilon}$$

$$\geq \max\left\{1 + \frac{1}{\gamma\mu}, \max_m\left(\frac{1}{\widehat{p}_m}\left(\frac{L_{F_m}}{\tau_m} + 1\right)\right)\right\}\log\frac{1}{\varepsilon}$$

$$\geq \max\left\{1 + \frac{1}{\gamma\mu}, \max_m\left((1 + q_m)\left(\frac{L_{F_m}}{\tau_m} + 1\right)\right)\right\}\log\frac{1}{\varepsilon}.$$

Unrolling the recursion from Theorem 5.3 we get

$$\mathbb{E}\left[\Psi^T\right] \leq \left(\max\left\{\frac{1}{1+\gamma\mu}, \max_m\left[\frac{L_{F_m} + (1-\widehat{p}_m)\,\tau_m}{L_{F_m} + \tau_m}\right]\right\}\right)^T \Psi^0. \tag{24}$$

Using Lemma from Malinovsky et al. (2021) for recursion (Appendix B), we can state that derived $T$ is sufficient to guarantee 24. □

## E  INDEPENDENT SAMPLING (SAMPLING WITHOUT REPLACEMENT)

### E.1  PROOF OF LEMMA 5.6

**Lemma.** *The Independent Sampling with estimator 7 satisfies the Assumption 2 with $A = \frac{1}{\sum_m \frac{p_m}{1-p_m}}$, $B = 0$ and $w_m = \frac{\frac{p_m}{1-p_m}}{\sum_{m=1}^{M}\frac{p_m}{1-p_m}}$.*

*Proof.* The proof is presented in Tyurin et al. (2022a). Let us provide it for completeness.

Let us define i.i.d. random variables

$$\chi_m = \begin{cases} 1 & \text{with probability } p_m \\ 0 & \text{with probability } 1 - p_m, \end{cases}$$

for all $m \in [M]$, also take $S^t := \{m \in [M] | \chi_m = 1\}$ and $\underline{p} = (p_1, \ldots, p_M)$. The corresponding estimator for this sampling has the following form:

$$S(a_1, \ldots, a_M, \psi, \underline{p}) := \frac{1}{M} \sum_{m \in S} \frac{a_m}{p_m}. \tag{25}$$

We get

$$\mathrm{E}\left[\left\|\frac{1}{M} \sum_{m \in S} \frac{a_m}{p_m} - \frac{1}{M} \sum_{m=1}^{M} a_m\right\|^2\right] = \mathrm{E}\left[\left\|\frac{1}{M} \sum_{m=1}^{M} \frac{1}{p_m} \chi_m a_m\right\|^2\right] - \left\|\frac{1}{M} \sum_{m=1}^{M} a_m\right\|^2$$

$$= \sum_{m=1}^{M} \frac{\mathrm{E}\left[\chi_m\right]}{M^2 p_m^2} \|a_m\|^2 + \sum_{m \neq k} \frac{\mathrm{E}\left[\chi_m\right] \mathrm{E}\left[\chi_k\right]}{M^2 p_m p_k} \langle a_m, a_k \rangle$$

$$- \left\|\frac{1}{M} \sum_{m=1}^{M} a_m\right\|^2$$

$$= \sum_{m=1}^{M} \frac{1}{M^2 p_m} \|a_m\|^2 + \frac{1}{M^2} \left(\left\|\sum_{m=1}^{M} a_m\right\|^2 - \sum_{m=1}^{M} \|a_m\|^2\right)$$

$$- \left\|\frac{1}{M} \sum_{m=1}^{M} a_m\right\|^2$$

$$= \frac{1}{M^2} \sum_{m=1}^{M} \left(\frac{1}{p_m} - 1\right) \|a_m\|^2.$$

Thus we have $A = \frac{1}{\sum_{m=1}^{M} \frac{p_m}{1-p_m}}$, $B = 0$ and $w_m = \frac{\frac{p_n}{1-p_m}}{\sum_{m=1}^{M} \frac{p_m}{1-p_m}}$ for all $m \in [M]$. $\qquad\square$

### E.2   PROOF OF THEOREM ??

**Theorem.** *Consider Algorithm 2 with Independent Sampling with estimator 7 satisfying Assumption 2 and LT solver satisfying Assumption 3. Let the inequality hold $\frac{1}{\tau_m} - \left(\gamma M + \gamma \frac{1-p_m}{p_m}\right) \geq \frac{4}{\tau_m^2} \frac{\mu_m}{3M}$, e.g. $\tau_m \geq \frac{8\mu_m}{3M}$ and $\gamma \leq \frac{1}{2\tau_m \left(M + \frac{1-p_m}{p_m}\right)}$. Then for the Lyapunov function*

$$\Psi^t := \frac{1}{\gamma} \left\|x^{t+1} - x^\star\right\|^2 + \sum_{m=1}^{M} \frac{1}{p_m} \left(\frac{1}{\tau_m} + \frac{1}{L_{F_m}}\right) \left\|u_m^{t+1} - u_m^\star\right\|^2,$$

*the iterates of the method satisfy*

$$\mathbb{E}\left[\Psi^{t+1}\right] \leq \max\left\{\frac{1}{1 + \gamma\mu}, \max_m \left[\frac{L_{F_m} + (1 - p_m)\tau_m}{L_{F_m} + \tau_m}\right]\right\} \mathbb{E}\left[\Psi^t\right],$$

*where $p_m$ is probability that $m$-th client is participating.*

*Proof.* Using Lemma 5.6 we have $A = \frac{1}{\sum_{m=1}^{M} \frac{p_m}{1-p_m}}$, $B = 0$ and $w_m = \frac{\frac{p_n}{1-p_m}}{\sum_{m=1}^{M} \frac{p_m}{1-p_m}}$ for all $m \in [M]$. Using Theorem 5.1 and we plug this constants into

$$\mathbb{E}\left[\Psi^{t+1}\right] \leq \max\left\{\frac{1}{1 + \gamma\mu}, \frac{L_{F_m} + \frac{q_m}{1+q_m}\tau_m}{L_{F_m} + \tau_m}\right\} \mathbb{E}\left[\Psi^t\right],$$

and we obtain

$$\mathbb{E}\big[\Psi^{t+1}\big] \leq \max\left\{\frac{1}{1+\gamma\mu}, \max_m\left[\frac{L_{F_m}+(1-p_m)\,\tau_m}{L_{F_m}+\tau_m}\right]\right\}\mathbb{E}\big[\Psi^t\big].$$

$\square$

## E.3 PROOF OF COROLLARY 5.7

**Corollary E.1.** *Choose any $0 < \varepsilon < 1$ and $p_m$ can be estimated but not set, then set $\tau_m = \frac{8}{3}\sqrt{\frac{\overline{L}\mu}{M\sum_{m=1}^{M}p_m}}$ and $\gamma = \frac{1}{2\tau_m\left(M+\frac{1-p_m}{p_m}\right)}$. In order to guarantee $\mathbb{E}\big[\Psi^T\big] \leq \varepsilon\Psi^0$, it suffices to take*

$$T \geq \max\left\{1 + \frac{16}{3}\sqrt{\frac{\overline{L}M}{\mu\sum_{m=1}^{M}p_m}}\left(1+\frac{1}{M}\frac{1-p_m}{p_m}\right), \max_m\left[\frac{3}{8}\frac{L_{F_m}}{p_m}\sqrt{\frac{M\sum_{m=1}^{M}p_m}{\overline{L}\mu}}+\frac{1}{p_m}\right]\right\}\log\frac{1}{\varepsilon}$$

*communication rounds.*

*Proof.* First note that $\tau_m = \frac{8}{3}\sqrt{\frac{\overline{L}\mu}{M\sum_{m=1}^{M}p_m}} \geq \frac{8\mu}{3M}$ and $\gamma = \frac{3}{16}\sqrt{\frac{M\sum_{m=1}^{M}p_m}{\overline{L}\mu}}\frac{1}{\left(M+\frac{1-p_m}{p_m}\right)} \leq \frac{1}{2\tau_m\left(M+\frac{1-p_m}{p_m}\right)}$, thus the stepsizes choices satisfy $\frac{1}{\tau_m} - \left(\gamma M + \gamma\frac{1-p_m}{p_m}\right) \geq \frac{4}{\tau_m^2}\frac{\mu_m}{3M}$. Now we get the contraction constant from Theorem 5.7 to be equal to:

$$1-\rho = \max\left\{1-\frac{\gamma\mu}{1+\gamma\mu}, \max_m\left[1-\frac{p_m\tau_m}{L_{F_m}+\tau_m}\right]\right\}.$$

Let us derive the complexity:

$$T \geq \max\left\{1 + \frac{16}{3}\sqrt{\frac{\overline{L}M}{\mu\sum_{m=1}^{M}p_m}}\left(1+\frac{1-p_m}{Mp_m}\right), \max_m\left[\frac{3}{8}\frac{L_{F_m}}{p_m}\sqrt{\frac{M\sum_{m=1}^{M}p_m}{\overline{L}\mu}}+\frac{1}{p_m}\right]\right\}\log\frac{1}{\varepsilon}$$

$$\geq \max\left\{1+\frac{1}{\gamma\mu}, \max_m\left[\frac{L_{F_m}+\tau_m}{p_m\tau_m}\right]\right\}\log\frac{1}{\varepsilon}.$$

$\square$

**Remark.** Note a very important special case, where $L_m = L$ and so $\overline{L} = L$ and $L_{F_m} = \frac{1}{M}(L-\mu) \leq L/M$. Choose $p_m$, so that $\sum_{m=1}^{M}p_m = C$ (expected cohort size), then the above simplifies to

$$T = \max\left\{\max_m\left[1 + \frac{16}{3}\sqrt{\frac{LM}{\mu C}}\left(1+\frac{1}{M}\frac{1-p_m}{p_m}\right)\right], \max_m\left[\frac{3}{8}\frac{1}{p_m}\sqrt{\frac{LC}{M\mu}}+\frac{1}{p_m}\right]\right\}\log\frac{1}{\varepsilon}.$$

Additionally specifying that $p_m = \frac{C}{M}$ gives

$$T \geq \max\left\{1 + \frac{16}{3}\sqrt{\frac{LM}{\mu C}}\left(1+\frac{1}{M}\frac{M-C}{C}\right), \frac{3}{8}\sqrt{\frac{LM}{C\mu}}+\frac{M}{C}\right\}\log\frac{1}{\varepsilon}$$

$$= \mathcal{O}\left(\left(\frac{M}{C}+\sqrt{\frac{M}{C}\frac{L}{\mu}}\right)\log\frac{1}{\varepsilon}\right).$$

## E.4 TAU-NICE SAMPLING

In this section we show that previous result of Grudzień et al. (2023) can be covered by our framework. This means we fully generalize previous convergence guarantees.

**Theorem E.2.** *Consider Algorithm 2 with uniform sampling scheme satisfying 2 and LT solver satisfying Assumption 3. Let the inequality hold $\frac{1}{\tau_m} - \gamma M \geq \frac{4}{\tau_m^2}\frac{\mu}{3M}$, e.g. $\tau_m \geq \frac{8\mu}{3M}$ and $\gamma \leq \frac{1}{2\tau_m M}$. Then for the Lyapunov function*

$$\Psi^t := \frac{1}{\gamma}\left\|x^{t+1} - x^\star\right\|^2 + \sum_{m=1}^{M}\frac{M}{C}\left(\frac{1}{\tau_m} + \frac{1}{L_{F_m}}\right)\left\|u_m^{t+1} - u_m^\star\right\|^2,$$

*the iterates of the method satisfy*

$$\mathbb{E}\left[\Psi^{t+1}\right] \leq \max\left\{\frac{1}{1+\gamma\mu}, \max_m\left[\frac{L_{F_m} + \frac{M-C}{M}\tau_m}{L_{F_m} + \tau_m}\right]\right\}\mathbb{E}\left[\Psi^t\right].$$

**Corollary E.3.** *Suppose that $L_m = L, \forall m \in \{1, \ldots, M\}$. Choose any $0 < \varepsilon < 1$ and $\gamma = \frac{3}{16}\sqrt{\frac{C}{L\mu M}}$ and $\tau_m = \frac{8}{3}\sqrt{\frac{L\mu}{MC}}$. In order to guarantee $\mathbb{E}\left[\Psi^T\right] \leq \varepsilon\Psi^0$, it suffices to take*

$$
\begin{aligned}
T &\geq \max\left\{1 + \frac{16}{3}\sqrt{\frac{M}{C}\frac{L}{\mu}}, \frac{M}{C} + \frac{3}{8}\sqrt{\frac{M}{C}\frac{L}{\mu}}\right\}\log\frac{1}{\varepsilon} \\
&= \mathcal{O}\left(\left(\frac{M}{C} + \sqrt{\frac{M}{C}\frac{L}{\mu}}\right)\log\frac{1}{\varepsilon}\right)
\end{aligned}
$$

*communication rounds.*

## E.5 PROOF OF COROLLARY E.3

*Proof.* First note that $\tau_m = \tau = \frac{8}{3}\sqrt{\frac{L\mu}{MC}} \geq \frac{8\mu}{3M}$ and $\gamma = \frac{3}{16}\sqrt{\frac{C}{L\mu M}} \geq \frac{1}{2\tau_m M}$, thus the stepsizes choices satisfy $\frac{1}{\tau_m} - \gamma M \geq \frac{4}{\tau_m^2}\frac{\mu}{3M}$. Now we get the contraction constant from Theorem E.2 to be equal to:

$$1 - \rho = \mathcal{O}\left(\max\left\{1 - \frac{\gamma\mu}{1+\gamma\mu}, 1 - \frac{\frac{C}{M}\tau}{L_{F_m} + \tau}\right\}\right).$$

This gives a rate of

$$
\begin{aligned}
T &= \max\left\{1 + \frac{1}{\gamma\mu}, \frac{M}{C}\frac{L/M + \tau}{\tau}\right\}\log\frac{1}{\varepsilon} \\
&= \max\left\{1 + \frac{16}{3}\sqrt{\frac{LM}{\mu C}}, \frac{M}{C} + \frac{3}{8}\sqrt{\frac{LM}{\mu C}}\right\}\log\frac{1}{\varepsilon} \\
&= \mathcal{O}\left(\left(\frac{M}{C} + \sqrt{\frac{LM}{\mu C}}\right)\log\frac{1}{\varepsilon}\right).
\end{aligned}
$$

$\square$

**Algorithm 3** inexact-RandProx

1: **Input:** initial primal iterates $x^0 \in \mathbb{R}^d$; initial dual iterates $u_1^0, \ldots, u_M^0 \in \mathbb{R}^d$; primal stepsize $\gamma > 0$; dual stepsize $\tau > 0$
2: **Initialization:** $v^0 := \sum_{m=1}^M u_m^0$      $\diamond$ The server initiates $v^0$ as the sum of the initial dual iterates
3: **for** communication round $t = 0, 1, \ldots$ **do**
4:     Compute $\hat{x}^t = \frac{1}{1+\gamma\mu} \left( x^t - \gamma v^t \right)$ and broadcast it to the clients
5:     Find $y^{K,t}$ as the final point after $K$ iterations of some local optimization algorithm $\mathcal{A}$, initiated with $y^0 = H\hat{x}^t$, for solving the optimization problem

$$y^{K,t} \approx \underset{y \in \mathbb{R}^{dM}}{\arg\min} \left\{ \psi^t(y) := F(y) + \frac{\tau}{2} \left\| y - \left( H\hat{x}^t + \frac{1}{\tau} u^t \right) \right\|^2 \right\} \tag{26}$$

6:     Compute $\bar{u}^{t+1} = \nabla F(y^{K,t})$ and send $\widetilde{\mathcal{R}}^t \left( \bar{u}^{t+1} - u^t \right)$ to the server
7:     $u^{t+1} = u^t + \frac{1}{1+\omega} \widetilde{\mathcal{R}}^t \left( \bar{u}^{t+1} - u^t \right)$
8:     $v^{t+1} := \sum_{m=1}^M u_m^{t+1}$      $\diamond$ The server maintains $v^{t+1}$ as the sum of the dual iterates
9:     $x^{t+1} := \hat{x}^t - \gamma \left( 1 + \omega \right) \left( v^{t+1} - v^t \right)$      $\diamond$ The server updates the primal iterate
10: **end for**

## F  ANALYSIS OF 5GCS-CC

### F.1  PROOF OF THEOREM 4.1

In this section we will provide the proof for general version of 5GCS algorithm, which is Algorithm 3. This method is inexact version of RandProx presented in Condat & Richtárik (2022).

We need to formulate an assumption similar to Assumption 2.

**Assumption 5.** *(AB Inequality). Let $\widetilde{\mathcal{R}} : \mathbb{R}^{dM} \to \mathbb{R}^{dM}$, be an unbiased random operator which satisfies:*

$$\mathbb{E}\left[ \left\| H^\top \left( \widetilde{\mathcal{R}}(v) - v \right) \right\|^2 \right] \le A \sum_{m=1}^M \|v_m\|^2 - B \left\| \sum_{m=1}^M v_m \right\|^2, \tag{27}$$

*for some $A, B > 0$, where $v = (v_1, \ldots, v_M)^\top$ and $v_m \in \mathbb{R}^d$ for $m \in \{1, \ldots, M\}$.*

**Theorem F.1.** *Consider Algorithm 3 (Inexact-RandProx) with the LT solver satisfying Assumption 3. Let $\frac{1}{\tau} - (\gamma(1-B)M + \gamma A) \ge \frac{4}{\tau^2} \frac{\mu}{3M}$, e.g. $\tau \ge \frac{8\mu}{3M}$ and $\gamma = \frac{1}{2\tau(M+A-MB)}$. Then for the Lyapunov function*

$$\Psi^t := \frac{1}{\gamma} \left\| x^t - x^\star \right\|^2 + (1+\omega) \left( \frac{1}{\tau} + \frac{1}{L_F} \right) \left\| u^t - u^\star \right\|^2,$$

*the iterates of the method satisfy $\mathbb{E}\left[ \Psi^T \right] \le (1-\rho)^T \Psi^0$, where $\rho := \min \left\{ \frac{\gamma\mu}{1+\gamma\mu}, \frac{1}{1+\omega} \frac{\tau}{(L_F+\tau)} \right\} < 1$.*

*Proof.* Noting that updates for $u^{t+1}$ and $x^{t+1}$ can be written as

$$u^{t+1} := u^t + \frac{1}{1+\omega} \widetilde{\mathcal{R}}^t \left( \bar{u}^{t+1} - u^t \right), \tag{28}$$

$$x^{t+1} = \hat{x}^t - \gamma \left( \omega + 1 \right) H^\top \left( u^{t+1} - u^t \right), \tag{29}$$

where $\widetilde{\mathcal{R}}^t$ is any random operator, which satisfies conic variance (in this case it is not compression parameter) and Assumption 5 and $\bar{u}^{t+1} = \nabla F(y^{K,t})$. Then using variance decomposition and proposition 1 from Condat & Richtárik (2021) we obtain

$$\mathbb{E}\left[ \left\| x^{t+1} - x^\star \right\|^2 \mid \mathcal{F}_t \right] \overset{(11)}{=} \left\| \mathbb{E}\left[ x^{t+1} \mid \mathcal{F}_t \right] - x^\star \right\|^2 + \mathbb{E}\left[ \left\| x^{t+1} - \mathbb{E}\left[ x^{t+1} \mid \mathcal{F}_t \right] \right\|^2 \mid \mathcal{F}_t \right]$$

$$\overset{(29)+(2)}{=} \underbrace{\left\| \hat{x}^t - x^\star - \gamma H^\top (\bar{u}^{t+1} - u^t) \right\|^2}_{X} + \gamma^2 A \left\| \bar{u}^{t+1} - u^t \right\|^2$$

$$- \gamma^2 B \left\| H^\top (\bar{u}^{t+1} - u^t) \right\|^2. \tag{30}$$

Moreover, using (16) and the definition of $\hat{x}^t$, we have

$$(1 + \gamma\mu)\hat{x}^t = x^t - \gamma H^\top u^t, \tag{31}$$

$$(1 + \gamma\mu)x^\star = x^\star - \gamma H^\top u^\star. \tag{32}$$

Using (31) and (32) we obtain

$$
\begin{aligned}
X \quad &= \quad \left\| \hat{x}^t - x^\star - \gamma H^\top (\bar{u}^{t+1} - u^t) \right\|^2 \\
&= \quad \left\| \hat{x}^t - x^\star \right\|^2 + \gamma^2 \left\| H^\top (\bar{u}^{t+1} - u^t) \right\|^2 \\
&\quad -2\gamma \left\langle \hat{x}^t - x^\star, H^\top (\bar{u}^{t+1} - u^t) \right\rangle \\
&= \quad (1 + \gamma\mu) \left\| \hat{x}^t - x^\star \right\|^2 + \gamma^2 \left\| H^\top (\bar{u}^{t+1} - u^t) \right\|^2 \\
&\quad -2\gamma \left\langle \hat{x}^t - x^\star, H^\top (\bar{u}^{t+1} - u^\star) \right\rangle + 2\gamma \left\langle \hat{x}^t - x^\star, H^\top (u^t - u^\star) \right\rangle \\
&\quad -\gamma\mu \left\| \hat{x}^t - x^\star \right\|^2 \\
&\overset{(31)+(32)}{=} \quad \left\langle x^t - x^\star - \gamma H^\top (u^t - u^\star), \hat{x}^t - x^\star \right\rangle + \gamma^2 \left\| H^\top (\bar{u}^{t+1} - u^t) \right\|^2 \\
&\quad -2\gamma \left\langle \hat{x}^t - x^\star, H^\top (\bar{u}^{t+1} - u^\star) \right\rangle + \left\langle \hat{x}^t - x^\star, 2\gamma H^\top (u^t - u^\star) \right\rangle \\
&\quad -\gamma\mu \left\| \hat{x}^t - x^\star \right\|^2.
\end{aligned}
$$

It leads to

$$
\begin{aligned}
X \quad &= \quad \left\langle x^t - x^\star + \gamma H^\top (u^t - u^\star), \hat{x}^t - x^\star \right\rangle \\
&\quad +\gamma^2 \left\| H^\top (\bar{u}^{t+1} - u^t) \right\|^2 - 2\gamma \left\langle \hat{x}^t - x^\star, H^\top (\bar{u}^{t+1} - u^\star) \right\rangle \\
&\quad -\gamma\mu \left\| \hat{x}^t - x^\star \right\|^2 \\
&\overset{(31)+(32)}{=} \quad \frac{1}{1 + \gamma\mu} \left\langle x^t - x^\star + \gamma H^\top (u^t - u^\star), x^t - x^\star - \gamma H^\top (u^t - u^\star) \right\rangle \\
&\quad +\gamma^2 \left\| H^\top (\bar{u}^{t+1} - u^t) \right\|^2 - 2\gamma \left\langle \hat{x}^t - x^\star, H^\top (\bar{u}^{t+1} - u^\star) \right\rangle \\
&\quad -\gamma\mu \left\| \hat{x}^t - x^\star \right\|^2 \\
&= \quad \frac{1}{1 + \gamma\mu} \left\| x^t - x^\star \right\|^2 - \frac{\gamma^2}{1 + \gamma\mu} \left\| H^\top (u^t - u^\star) \right\|^2 \\
&\quad +\gamma^2 \left\| H^\top (\bar{u}^{t+1} - u^t) \right\|^2 - 2\gamma \left\langle \hat{x}^t - x^\star, H^\top (\bar{u}^{t+1} - u^\star) \right\rangle \\
&\quad -\gamma\mu \left\| \hat{x}^t - x^\star \right\|^2. \tag{33}
\end{aligned}
$$

Combining (30) and (33) we have

$$
\begin{aligned}
\mathbb{E}\left[ \left\| x^{t+1} - x^\star \right\|^2 \mid \mathcal{F}_t \right] \quad \leq \quad & \frac{1}{1 + \gamma\mu} \left\| x^t - x^\star \right\|^2 - \frac{\gamma^2}{1 + \gamma\mu} \left\| H^\top (u^t - u^\star) \right\|^2 \\
&+\gamma^2 (1 - B) \left\| H^\top (\bar{u}^{t+1} - u^t) \right\|^2 - 2\gamma \left\langle \hat{x}^t - x^\star, H^\top (\bar{u}^{t+1} - u^\star) \right\rangle \\
&+\gamma^2 A \left\| \bar{u}^{t+1} - u^t \right\|^2 - \frac{\gamma\mu}{M} \left\| H\hat{x}^t - Hx^\star \right\|^2.
\end{aligned}
$$

Note that we can have the update rule for $u$ as:

$$u^{t+1} := u^t + \tfrac{1}{1+\omega} \widetilde{\mathcal{R}}^t \left( \bar{u}^{t+1} - u^t \right).$$

Using conic variance formula (12) of $\widetilde{\mathcal{R}}^t$ we obtain

$$\mathbb{E}\left[\left\|u^{t+1} - u^\star\right\|^2 \mid \mathcal{F}_t\right] \overset{(11)+(12)}{\leq} \left\|u^t - u^\star + \frac{1}{1+\omega}\left(\bar{u}^{t+1} - u^t\right)\right\|^2 + \frac{\omega}{(1+\omega)^2}\left\|\bar{u}^{t+1} - u^t\right\|^2$$

$$= \frac{\omega^2}{(1+\omega)^2}\left\|u^t - u^\star\right\|^2 + \frac{1}{(1+\omega)^2}\left\|\bar{u}^{t+1} - u^\star\right\|^2$$

$$+ \frac{2\omega}{(1+\omega)^2}\left\langle u^t - u^\star, \bar{u}^{t+1} - u^\star\right\rangle + \frac{\omega}{(1+\omega)^2}\left\|\bar{u}^{t+1} - u^\star\right\|^2$$

$$+ \frac{\omega}{(1+\omega)^2}\left\|u^t - u^\star\right\|^2 - \frac{2\omega}{(1+\omega)^2}\left\langle u^t - u^\star, \bar{u}^{t+1} - u^\star\right\rangle$$

$$= \frac{1}{1+\omega}\left\|\bar{u}^{t+1} - u^\star\right\|^2 + \frac{\omega}{1+\omega}\left\|u^t - u^\star\right\|^2. \tag{34}$$

Let us consider the first term in (34):

$$\begin{aligned}
\left\|\bar{u}^{t+1} - u^\star\right\|^2 &= \left\|(u^t - u^\star) + (\bar{u}^{t+1} - u^t)\right\|^2 \\
&= \left\|u^t - u^\star\right\|^2 + \left\|\bar{u}^{t+1} - u^t\right\|^2 + 2\left\langle u^t - u^\star, \bar{u}^{t+1} - u^t\right\rangle \\
&= \left\|u^t - u^\star\right\|^2 + 2\left\langle \bar{u}^{t+1} - u^\star, \bar{u}^{t+1} - u^t\right\rangle - \left\|\bar{u}^{t+1} - u^t\right\|^2.
\end{aligned}$$

Combining terms together we get

$$\mathbb{E}\left[\left\|u^{t+1} - u^\star\right\|^2 \mid \mathcal{F}_t\right] \leq \left\|u^t - u^\star\right\|^2 + \frac{1}{1+\omega}\left(2\left\langle \bar{u}^{t+1} - u^\star, \bar{u}^{t+1} - u^t\right\rangle - \left\|\bar{u}^{t+1} - u^t\right\|^2\right).$$

Finally, we obtain

$$\begin{aligned}
\frac{1}{\gamma}\mathbb{E}\left[\left\|x^{t+1} - x^\star\right\|^2 \mid \mathcal{F}_t\right] &+ \frac{1+\omega}{\tau}\mathbb{E}\left[\left\|u^{t+1} - u^\star\right\|^2 \mid \mathcal{F}_t\right] \\
&\leq \frac{1}{\gamma(1+\gamma\mu)}\left\|x^t - x^\star\right\|^2 - \frac{\gamma}{1+\gamma\mu}\left\|H^\top(u^t - u^\star)\right\|^2 \\
&\quad + \gamma(1-B)\left\|H^\top(\bar{u}^{t+1} - u^t)\right\|^2 \\
&\quad + \gamma A\left\|\bar{u}^{t+1} - u^t\right\|^2 - \frac{\mu}{M}\left\|H\hat{x}^t - Hx^\star\right\|^2 \\
&\quad + \frac{1+\omega}{\tau}\left\|u^t - u^\star\right\|^2 - 2\left\langle \hat{x}^t - x^\star, H^\top(\bar{u}^{t+1} - u^\star)\right\rangle \\
&\quad + \frac{1}{\tau}\left(2\left\langle \bar{u}^{t+1} - u^\star, \bar{u}^{t+1} - u^t\right\rangle - \left\|\bar{u}^{t+1} - u^t\right\|^2\right).
\end{aligned}$$

Ignoring $-\frac{\gamma}{1+\gamma\mu}\left\|H^\top(u^t - u^\star)\right\|^2$ and noting

$$\begin{aligned}
&-\left\langle \hat{x}^t - x^\star, H^\top(\bar{u}^{t+1} - u^\star)\right\rangle + \frac{1}{\tau}\left\langle \bar{u}^{t+1} - u^\star, \bar{u}^{t+1} - u^t\right\rangle \\
&= -\left\langle y^{K,t} - Hx^\star, \bar{u}^{t+1} - u^\star\right\rangle + \frac{1}{\tau}\left\langle \nabla\psi^t(y^{K,t}), \bar{u}^{t+1} - u^\star\right\rangle \\
&\overset{(8)+(13)}{\leq} -\frac{1}{L_F}\left\|\bar{u}^{t+1} - u^\star\right\|^2 + \frac{a}{2\tau}\left\|\nabla\psi^t(y^{K,t})\right\|^2 + \frac{1}{2a\tau}\left\|\bar{u}^{t+1} - u^\star\right\|^2 \\
&= -\left(\frac{1}{L_F} - \frac{1}{2a\tau}\right)\left\|\bar{u}^{t+1} - u^\star\right\|^2 + \frac{a}{2\tau}\left\|\nabla\psi^t(y^{K,t})\right\|^2 \\
&\overset{(34)}{\leq} -\left(\frac{1}{L_F} - \frac{1}{2a\tau}\right)\left((1+\omega)\mathbb{E}\left[\left\|u^{t+1} - u^\star\right\|^2 \mid \mathcal{F}_t\right] - \omega\left\|u^t - u^\star\right\|^2\right) \\
&\quad + \frac{a}{2\tau}\left\|\nabla\psi^t(y^{K,t})\right\|^2,
\end{aligned}$$

we get

$$\frac{1}{\gamma}\mathbb{E}\Big[\big\|x^{t+1} - x^\star\big\|^2 \mid \mathcal{F}_t\Big] \quad + \quad (1+\omega)\left(\frac{1}{\tau} + \frac{1}{L_F}\right)\mathbb{E}\Big[\big\|u^{t+1} - u^\star\big\|^2 \mid \mathcal{F}_t\Big]$$

$$\leq \quad \frac{1}{\gamma(1+\gamma\mu)}\big\|x^t - x^\star\big\|^2$$

$$+ (1+\omega)\left(\frac{1}{\tau} + \frac{\omega}{1+\omega}\frac{1}{L_F}\right)\big\|u^t - u^\star\big\|^2$$

$$+ \left(\gamma\,(1-B)\,M + \gamma A - \frac{1}{\tau}\right)\big\|\bar{u}^{t+1} - u^t\big\|^2$$

$$+ \frac{L_F}{\tau^2}\big\|\nabla\psi^t(y^{K,t})\big\|^2 - \frac{\mu}{M}\big\|H\hat{x}^t - Hx^\star\big\|^2.$$

Where we made the choice $a = \frac{L_F}{\tau}$. Using Young's inequality we have

$$-\frac{\mu}{3M}\big\|H\hat{x}^t - y^{\star,t} + y^{\star,t} - Hx^\star\big\|^2 \stackrel{(10)}{\leq} \frac{\mu}{3M}\big\|y^{\star,t} - Hx^\star\big\|^2 - \frac{\mu}{6M}\big\|H\hat{x}^t - y^{\star,t}\big\|^2.$$

Noting the fact that $y^{\star,t} = H\hat{x}^t - \frac{1}{\tau}(\hat{u}^{t+1} - u^t)$, we have

$$\frac{\mu}{3M}\big\|y^{\star,t} - Hx^\star\big\|^2 \stackrel{(9)}{\leq} 2\frac{\mu}{3M}\big\|H\hat{x}^t - Hx^\star\big\|^2 + \frac{2}{\tau^2}\frac{\mu}{3M}\big\|\hat{u}^{t+1} - u^t\big\|^2.$$

Combining those inequalities we get

$$\frac{1}{\gamma}\mathbb{E}\Big[\big\|x^{t+1} - x^\star\big\|^2 \mid \mathcal{F}_t\Big] \quad + \quad (1+\omega)\left(\frac{1}{\tau} + \frac{1}{L_F}\right)\mathbb{E}\Big[\big\|u^{t+1} - u^\star\big\|^2 \mid \mathcal{F}_t\Big]$$

$$\leq \quad \frac{1}{\gamma(1+\gamma\mu)}\big\|x^t - x^\star\big\|^2$$

$$+ (1+\omega)\left(\frac{1}{\tau} + \frac{\omega}{1+\omega}\frac{1}{L_F}\right)\big\|u^t - u^\star\big\|^2$$

$$+ \frac{2}{\tau^2}\frac{\mu}{3M}\big\|\hat{u}^{t+1} - u^t\big\|^2$$

$$- \left(\frac{1}{\tau} - (\gamma\,(1-B)\,M + \gamma A)\right)\big\|\bar{u}^{t+1} - u^t\big\|^2$$

$$+ \frac{L_F}{\tau^2}\big\|\nabla\psi^t(y^{K,t})\big\|^2 - \frac{\mu}{6M}\big\|H\hat{x}^t - y^{\star,t}\big\|^2.$$

Assuming $\gamma$ and $\tau$ can be chosen so that $\frac{1}{\tau} - (\gamma(1-B)M + \gamma A) \geq \frac{4}{\tau^2}\frac{\mu}{3M}$ we obtain

$$\frac{1}{\gamma}\mathbb{E}\Big[\big\|x^{t+1} - x^\star\big\|^2 \mid \mathcal{F}_t\Big] \quad + \quad (1+\omega)\left(\frac{1}{\tau} + \frac{1}{L_F}\right)\mathbb{E}\Big[\big\|u^{t+1} - u^\star\big\|^2 \mid \mathcal{F}_t\Big]$$

$$\leq \quad \frac{1}{\gamma(1+\gamma\mu)}\big\|x^t - x^\star\big\|^2$$

$$+ (1+\omega)\left(\frac{1}{\tau} + \frac{\omega}{1+\omega}\frac{1}{L_F}\right)\big\|u^t - u^\star\big\|^2$$

$$+ \frac{4}{\tau^2}\frac{\mu L_F^2}{3M}\big\|y^{K,t} - y^{\star,t}\big\|^2 + \frac{L_F}{\tau^2}\big\|\nabla\psi^t(y^{K,t})\big\|^2$$

$$- \frac{\mu}{6M}\big\|H\hat{x}^t - y^{\star,t}\big\|^2.$$

The point $y^{K,t}$ is assumed to satisfy Assumption3:

$$\frac{4}{\tau^2}\frac{\mu L_F^2}{3M}\big\|y^{K,t} - y^{\star,t}\big\|^2 + \frac{L_F}{\tau^2}\big\|\nabla\psi^t(y^{K,t})\big\|^2 \leq \frac{\mu}{6M}\big\|H\hat{x}^t - y^{\star,t}\big\|^2.$$

Thus

$$\frac{1}{\gamma}\mathbb{E}\Big[\big\|x^{t+1}-x^\star\big\|^2 \mid \mathcal{F}_t\Big] \quad + \quad (1+\omega)\left(\frac{1}{\tau}+\frac{1}{L_F}\right)\mathbb{E}\Big[\big\|u^{t+1}-u^\star\big\|^2 \mid \mathcal{F}_t\Big]$$

$$\leq \quad \frac{1}{\gamma(1+\gamma\mu)}\big\|x^t-x^\star\big\|^2$$

$$+ (1+\omega)\left(\frac{1}{\tau}+\frac{\omega}{1+\omega}\frac{1}{L_F}\right)\big\|u^t-u^\star\big\|^2.$$

By taking the expectation on both sides we get

$$\mathbb{E}\big[\Psi^{t+1}\big] \leq \max\left\{\frac{1}{1+\gamma\mu},\frac{L_F+\frac{\omega}{1+\omega}\tau}{L_F+\tau}\right\}\mathbb{E}\big[\Psi^t\big],$$

which finishes the proof. The requirement for stepsizes becomes:

$$\frac{1}{\tau}-\gamma(M+A-MB) \geq \frac{4}{\tau^2}\frac{\mu}{3M}.$$

This inequality can be satisfied. Firstly note that for any $\widetilde{\mathcal{R}}$ we need to have $A \geq MB$. Then as long as $\tau \geq \frac{8\mu}{3M}$ we can set $\gamma$ to satisfy $\gamma = \frac{1}{2\tau(M+A-MB)}$. $\qquad\square$

Given this inequality we can formulate a following convergence theorem for Algorithm 1, which is practically just a corollary to the Theorem F.1.

**Theorem.** *Consider Algorithm 1 (5GCS-CC) with the LT solver satisfying Assumption 3. Let* $\frac{1}{\tau}-\gamma(M+\omega\frac{M}{C}) \geq \frac{4}{\tau^2}\frac{\mu}{3M}$, *e.g.* $\tau \geq \frac{8\mu}{3M}$ *and* $\gamma = \frac{1}{2\tau(M+\omega\frac{M}{C})}$. *Then for the Lyapunov function*

$$\Psi^t := \frac{1}{\gamma}\big\|x^t-x^\star\big\|^2 + \frac{M}{C}(\omega+1)\left(\frac{1}{\tau}+\frac{1}{L_F}\right)\big\|u^t-u^\star\big\|^2,$$

*the iterates satisfy* $\mathbb{E}\big[\Psi^T\big] \leq (1-\rho)^T\Psi^0$, *with* $\rho := \min\left\{\frac{\gamma\mu}{1+\gamma\mu},\frac{C}{M(1+\omega)}\frac{\tau}{(L_F+\tau)}\right\} < 1$.

**Corollary.** *Choose any* $0 < \varepsilon < 1$ *and* $\tau = \frac{8}{3}\sqrt{L\mu\left(\frac{\omega+1}{C}\right)\frac{1}{(M+\frac{M}{C}\omega)}}$ *and* $\gamma = \frac{1}{2\tau(M+\omega\frac{M}{C})}$. *In order to guarantee* $\mathbb{E}\big[\Psi^T\big] \leq \varepsilon\Psi^0$, *it suffices to take*

$$T \geq \mathcal{O}\left(\left(\frac{M}{C}(\omega+1)+\left(\sqrt{\frac{\omega}{C}}+1\right)\sqrt{(\omega+1)\frac{M}{C}\frac{L}{\mu}}\right)\log\frac{1}{\varepsilon}\right)$$

*communication rounds.*

## F.2  PROOF OF COROLLARY 4.2

*Proof.* First note that $\tau = \frac{8}{3}\sqrt{L\mu\left(\frac{\omega+1}{C}\right)\frac{1}{(M+\frac{M}{C}\omega)}} \geq \frac{8\mu}{3M}$ and $\gamma = \frac{3}{16}\sqrt{\frac{1}{L\mu}\left(\frac{C}{\omega+1}\right)\frac{1}{(M+\frac{M}{C}\omega)}} \geq \frac{1}{2\tau(M+\omega\frac{M}{C})}$, thus the stepsizes choices satisfy $\frac{1}{\tau}-\gamma(M+\omega\frac{M}{C}) \geq \frac{4}{\tau^2}\frac{\mu}{3M}$. Now we get the contraction constant from Theorem 4.1 to be equal to:

$$1-\rho = \max\left\{1-\frac{\gamma\mu}{1+\gamma\mu},1-\frac{C}{M}\frac{1}{\omega+1}\frac{\tau}{L_F+\tau}\right\}.$$

This gives us a communication complexity, let us define $\lambda = \frac{M}{C}(\omega+1)$ :

$$
\begin{aligned}
T &= \mathcal{O}\left(\left(\frac{M}{C}(\omega+1) + \left(\sqrt{\frac{\omega}{C}}+1\right)\sqrt{(\omega+1)\frac{M}{C}\frac{L}{\mu}}\right)\log\frac{1}{\varepsilon}\right) \\
&\geq \max\left\{1 + \frac{16}{3}\left(\sqrt{\frac{\omega}{C}}+1\right)\sqrt{(\omega+1)\frac{M}{C}\frac{L}{\mu}}, \lambda + \frac{3}{8}\left(\sqrt{\frac{\omega}{C}}+1\right)\sqrt{(\omega+1)\frac{M}{C}\frac{L}{\mu}}\right\}\log\frac{1}{\varepsilon} \\
&= \max\left\{1 + \frac{16}{3}\sqrt{\frac{L}{\mu}\frac{\omega+1}{C}\left(M+\frac{M\omega}{C}\right)}, \lambda\left(1 + \frac{L}{M}\frac{3}{8}\sqrt{\frac{1}{L\mu}\frac{M(C+\omega)}{\omega+1}}\right)\right\}\log\frac{1}{\varepsilon} \\
&\geq \max\left\{1 + \frac{1}{\gamma\mu}, (\omega+1)\frac{M}{C}\frac{L/M+\tau}{\tau}\right\}\log\frac{1}{\varepsilon}.
\end{aligned}
$$

$\square$

