# OpenReview forum: "Improving Accelerated Federated Learning with Compression and Importance Sampling"
_ICLR.cc/2024/Conference — Submitted to ICLR 2024_

### Official Review · Reviewer_AQbc · 2023-10-21

**Soundness:** 2 fair
**Presentation:** 2 fair
**Contribution:** 2 fair
**Rating:** 3
**Confidence:** 3

**Summary:**

The paper studied the communication problem in the federated learning. The authors proposed a local training strategy which combine the communication compression and importance client sampling. The authors provided theoretical analysis to validate their proposed methods.

**Strengths:**

The authors introduced an innovative federated training method that integrates client importance sampling. Additionally, the paper offers a thorough and comprehensive theoretical analysis of their approach.

**Weaknesses:**

The paper appears to encompass a broad range of topics, making it challenging to follow. The theoretical analysis is primarily confined to strongly convex scenarios, which may not be applicable to intricate models such as deep neural networks (DNNs). Moreover, the simplicity of the numerical experiments detracts from their overall persuasiveness. I would suggest to reorganize the presentation and include more experiments.

**Questions:**

1. In Algorithm 1&2 line 7, what if we can not optimize the problem in K iteration? How would the optimization error on  Eq(4) or Eq(6)  affect the final convergence? Has this be considered in Theorem 4.1 or 5.1?

2. How will the client's data heterogeneity affect the convergence as well as the sampling mechanism?

3. The assumption 2 seems not to be very clear. Can you explain how to achieve it in the algorithm?

4. The client sampling mechanism seems to be unclear. Is it informative? Which information is leveraged for the sampling?

5. Could the results be extended to non-convex cases? I would suggest to both include analysis and experiment on non-convex learning problems.

---

> ### Author Response · Authors · 2023-11-20
> **Reply to Review**
>
> **Strengths:**
>
> >The authors introduced an innovative federated training method that integrates client importance sampling. Additionally, the paper offers a thorough and comprehensive theoretical analysis of their approach.
>
> Thank you for providing your valuable feedback! Your insights and comments are greatly appreciated as they contribute to the ongoing refinement and improvement of our work. If you have any further thoughts, suggestions, or questions, please feel free to share them. Your engagement is instrumental in the continued development of our project, and we look forward to any additional input you may have. Once again, thank you for taking the time to offer your feedback.
>
> **Weaknesses:**
>
> >The paper appears to encompass a broad range of topics, making it challenging to follow. The theoretical analysis is primarily confined to strongly convex scenarios, which may not be applicable to intricate models such as deep neural networks (DNNs). Moreover, the simplicity of the numerical experiments detracts from their overall persuasiveness. I would suggest to reorganize the presentation and include more experiments.
>
> Thank you for dedicating your time and effort to providing comments. Allow us to address the weakness section first. We acknowledge the reviewer's observation that our method may not be directly applicable to optimize Deep Neural Networks (DNNs). This limitation arises due to the non-convex nature of the optimization landscape in DNNs, while our theoretical assumption necessitates strong convexity. Despite this, we believe the identified limitation holds significance. Our work introduces two novel algorithms, each designed to address critical challenges in Federated Learning, specifically compression and arbitrary sampling. Both methods achieve accelerated convergence rates by leveraging a widely used mechanism—local training.
>
> We wish to express our respectful disagreement with the viewpoint suggesting that the current landscape of Federated Learning is solely concentrated on Deep Neural Networks (DNNs). In reality, Federated Learning finds its predominant application in on-device Machine Learning environments, where each device is often constrained by limited computational resources. This inherent limitation implies that smaller devices may struggle to accommodate the training and execution demands associated with large-scale models like DNNs.
>
> Moreover, it's important to highlight that logistic regression, despite its simplicity, remains one of the most popular models in practical applications, especially within the context of mobile devices. The practicality and efficiency of logistic regression make it a preferred choice, particularly when considering the resource constraints often inherent in on-device Machine Learning settings. Therefore, we believe it is essential to recognize the diversity of models and applications within the Federated Learning paradigm, acknowledging the prevalent use of simpler models like logistic regression in real-world scenarios.
>
> It is important to note that the theoretical analysis of this mechanism, local training, is still in its early stages of development. Consequently, we aim to contribute to this evolving field by providing a concise summary of the generations of local training methods in our work. We consider our contribution to be novel in its analytical approach, as it addresses and resolves important problems while offering a comprehensive overview of local training methods.
>
> While we acknowledge that the experiments in our work are relatively straightforward, we assert that the paper's primary value lies in its theoretical results. The experiments serve the purpose of visually illustrating these results rather than constituting standalone findings. We appreciate your consideration of these aspects in evaluating the overall merit of our work.

---

> ### Author Response · Authors · 2023-11-20
> **Reply to Review 2**
>
> **Questions:**
> >1. In Algorithm 1&2 line 7, what if we can not optimize the problem in K iteration? How would the optimization error on Eq(4) or Eq(6) affect the final convergence? Has this be considered in Theorem 4.1 or 5.1?
>
> In the event that we fail to meet the Local Training assumption, it becomes challenging to assert that we will theoretically converge at the promised speeds. The potential arises for certain clients to address their local optimization problems more effectively than others, possibly employing distinct optimization strategies that result in varying convergence rates. Given that the Local Training Assumption is global, there exists the possibility of still satisfying it even if not every device can effectively solve its respective optimization problem.
>
> >2. How will the client's data heterogeneity affect the convergence as well as the sampling mechanism?
>
> We tackle the challenge of client data heterogeneity through our innovative algorithm, 5GCS-AB. This algorithm enables the utilization of sampling techniques, such as Importance Sampling (IS), which specifically accounts for the diversity present in the data. In contrast to other Local Training methods that mandate the use of upper and lower bounds on L-smoothness and mu-convexity of the functions, often resulting in substantial slowdowns in convergence rates, 5GCS-AB stands out. It accommodates arbitrary differences in the data while still achieving the optimal convergence rate.
>
> >3. The assumption 2 seems not to be very clear. Can you explain how to achieve it in the algorithm?
>
> Assumption 2 establishes a category of sampling schemes that encompasses a broad range of methods commonly employed in practical applications. This classification is designed to be inclusive, covering most existing sampling schemes that are prevalent in real-world scenarios. As part of our analytical framework, we possess the ability to rigorously examine and verify whether a given sampling scheme adheres to the conditions outlined in Assumption 2 and, subsequently, determine its inclusion within this specified class of samplings. This approach facilitates a systematic evaluation of the suitability of various sampling techniques within the framework of our analysis.
>
> >4. The client sampling mechanism seems to be unclear. Is it informative? Which information is leveraged for the sampling?
>
> The client sampling mechanism delineated in our approach exhibits a high degree of generality, offering substantial flexibility in its application. This versatility is further underscored by the weighted AB assumption, which introduces a nuanced level of adaptability to accommodate various scenarios and requirements. The weighted AB assumption allows for the incorporation of different weights, providing a framework that can be tailored to specific contexts, thereby enhancing the adaptability and applicability of the mechanism.
>
> In order to employ Importance Sampling effectively, it is imperative to possess information about the individual smoothness constants denoted as $L_i$. These constants play a crucial role in guiding the sampling mechanism, allowing for a targeted and informed approach to variance reduction during the optimization process. The incorporation of individual smoothness constants enhances the precision and adaptability of Importance Sampling, contributing to more efficient and effective optimization strategies tailored to the specific characteristics of each component in the system.
>
> >5. Could the results be extended to non-convex cases? I would suggest to both include analysis and experiment on non-convex learning problems.
>
> Our ambition is to broaden the scope of our acceleration methods to encompass non-convex scenarios. However, achieving this goal requires a fundamentally different analysis and approach, which is presently absent from both our knowledge and the existing literature. While our proofs successfully demonstrate convergence concerning the distance to the optimum—a distinctive feature in cases like ours—it is imperative to recognize that such a claim is not universally applicable.
>
> In the context of non-convex optimization, the intricacies of the landscape necessitate a novel perspective and analytical framework that currently eludes us. The proofs we provide, grounded in the specifics of convex optimization, cannot be seamlessly extended to the non-convex case due to the inherent disparities in optimization dynamics. As we strive to advance our understanding and methodologies, addressing the challenges posed by non-convex scenarios remains an avenue for future exploration and research.

---

> > ### Comment · Reviewer_AQbc · 2023-11-21
> > **Thanks for your response.**
> >
> > I would like to thank authors for the detailed response. However, my primary concerns for the paper are not addressed. I would suggest the authors to further elaborate:
> >
> > 1. Most of the optimization problem cannot be solved with limited iteration (i.e. K iterations). From this point, can the global theoretical results be based on the precision of the sub-problem?
> >
> > 2. I agree that the IS can help handle the data heterogeneity issue. But the proposed 5GCS-AB algorithm doesn't show how to set up the IS based on the heterogeneity.
> >
> > 3. Can you provided a few examples which satisfies the Assumption 2?
> >
> > 4. The current study and analysis are only on the strongly convex case. But Almost of the real-world problems are non-convex, especially under FL context. This limits this paper's contribution.
> >
> > I feel that there are many aspects of this article that still need improvement, including but not limited to the completeness of the theory, the connection between theory and real-world scenarios, reasonable assumptions, and more empirical evidence. Thus, I would keep my current score.

---

> > > ### Author Response · Authors · 2023-11-22
> > > **Reply to comment 1**
> > >
> > > Thank you for your comment!
> > >
> > > > 1. Most of the optimization problem cannot be solved with limited iteration (i.e. K iterations). From this point, can the global theoretical results be based on the precision of the sub-problem?
> > >
> > > To elaborate further, our approach centers around the careful consideration of a specific sub-problem that we have purposefully constructed to align with the requirements of our method. This sub-problem is characterized by advantageous properties, notably strong convexity and L-smoothness, which contribute to the effectiveness of our overall methodology.
> > >
> > > It is worth noting that our methodology does not necessitate an exact solution to the aforementioned sub-problem. Instead, we find that obtaining an approximate solution suffices for our purposes. Importantly, this approximate solution can be achieved within a constrained number of iterations, underscoring the efficiency of our approach.
> > >
> > > The generality of our analysis is a key strength. We have established a condition wherein the accurate resolution of the local sub-problem lends itself to seamless integration into various methods. This adaptability enhances the applicability of our approach across diverse contexts and problem domains.
> > >
> > > Moreover, our investigation delves into the nuanced relationship between the number of local steps and the corresponding number of communication rounds. This intricate analysis is particularly pertinent in the context of gradient descent, as detailed in the 5GCS paper. By comprehensively exploring this interplay, we contribute valuable insights that shed light on the dynamics of our method.
> > >
> > > Grudzień, M., Malinovsky, G., & Richtárik, P. (2023, April). Can 5th Generation Local Training Methods Support Client Sampling? Yes!. In International Conference on Artificial Intelligence and Statistics (pp. 1055-1092). PMLR.
> > >
> > >  >2. I agree that the IS can help handle the data heterogeneity issue. But the proposed 5GCS-AB algorithm doesn't show how to set up the IS based on the heterogeneity.
> > >
> > > We respectfully express our disagreement with the aforementioned statement. Heterogeneity is a distinguishing characteristic marked by significant variations in the smoothness constants, denoted as $L_i$. The importance sampling strategy allocates different probabilities for client participation, contingent upon their respective smoothness constants, as indicated by the formula:
> > >
> > > $$ p_m = \frac{\sqrt{L_m}}{\sum_{m=1}^M \sqrt{L_m}}. $$
> > >
> > > It is evident that these probabilities are contingent upon the individual smoothness constants, emphasizing the influence of each $L_i$. Furthermore, the significance of a particular $L_i$ is accentuated when compared to other smoothness constants. It is pertinent to note that in cases where all individual smoothness constants are identical (homogeneous scenario), the optimal probabilities converge to uniform distribution: $ p_m = \frac{1}{M} $. This implies an equal and uniform distribution of participation.
> > >
> > > We firmly assert that the setup of Importance Sampling is intricately tied to the concept of heterogeneity. If additional clarification is required, we are more than willing to provide further explanations.

---

> > > > ### Comment · Reviewer_AQbc · 2023-11-22
> > > >
> > > > 1. `our approach centers around the careful consideration of a specific sub-problem that we have purposefully constructed to align with the requirements of our method` -- If the algorithm only targets a sub-set of optimization problem, then the contribution would be limited. Strongly convexity and L-smoothness would be some general assumption to capture the objection function's landscape but identifying a solution with limited iteration would be unrealistic assumption on the algorithm.
> > > >
> > > > 2. `It is worth noting that our methodology does not necessitate an exact solution to the aforementioned sub-problem. Instead, we find that obtaining an approximate solution suffices for our purposes.` -- I agree with this point and that's why I encourage the authors to improve the theoretical results by considering the precision of the solution of subproblem.
> > > >
> > > > 3. `Heterogeneity is a distinguishing characteristic marked by significant variations in the smoothness constants` -- I disagree with this statement. Heterogeneity will definitely affect but not just the smoothness. Take LSE problem as an example $E_{(y,X)~P_i}\|y- X\beta\|^2$. The heterogeneity of $X$ will affect the smoothness and convexity and that of $y$ will affect the location of solution.
> > > >
> > > > 4. Even though we only consider the heterogeneity affects the smoothness. $L_i$ for each local loss might be hard to capture.

---

> > > > > ### Author Response · Authors · 2023-11-22
> > > > > **Reply**
> > > > >
> > > > > >If the algorithm only targets a sub-set of optimization problem, then the contribution would be limited. Strongly convexity and L-smoothness would be some general assumption to capture the objection function's landscape but identifying a solution with limited iteration would be unrealistic assumption on the algorithm.
> > > > >
> > > > > We appreciate the opportunity to clarify our position on these statements. It seems there might be a misunderstanding. As algorithm designers, we intentionally **construct** a sub-problem to be smooth and strongly convex. It's crucial to emphasize that this is **NOT** an assumption; rather, we take deliberate measures to ensure that the sub-problem is $L$-smooth and strongly convex. We achieve this by **designing** the sub-problem to be $L_{F_m}+\tau_m$ smooth and $\tau_m$ strongly convex, and we exercise control over these parameters using dual step sizes $\tau_m.$
> > > > >
> > > > > The guarantee we provide is that any method capable of converging to a solution for a smooth and strongly convex sub-problem—of which there are numerous existing methods—can fulfill Assumption 3. In essence, Assumption 3 serves as a stoppage criterion, enabling the calculation of the number of local steps.
> > > > >
> > > > > In summary, the smoothness and strong convexity of the sub-problem are **NOT** assumed; they are deliberately incorporated into the **design**. Any method proficient in handling strongly convex and smooth functions meets Condition 3 and is compatible with our algorithm.
> > > > >
> > > > > >I agree with this point and that's why I encourage the authors to improve the theoretical results by considering the precision of the solution of subproblem.
> > > > >
> > > > > A comprehensive analysis of the relationship between the precision of the solution and the number of communication rounds has been conducted in the following work:
> > > > >
> > > > > Grudzień, M., Malinovsky, G., & Richtárik, P. (2023, April). Can 5th Generation Local Training Methods Support Client Sampling? Yes!. In International Conference on Artificial Intelligence and Statistics (pp. 1055-1092). PMLR.
> > > > >
> > > > > Utilizing established findings holds little interest for us and falls outside the scope of our consideration, as it fails to offer novel insights.
> > > > >
> > > > > >I disagree with this statement. Heterogeneity will definitely affect but not just the smoothness...
> > > > >
> > > > > Let us clarify several aspects. Firstly, it's important to note that at no point in the text or discussion do we claim that heterogeneity affects **only** smoothness. In the context of Federated Optimization, where the loss function follows a finite sum structure,
> > > > >
> > > > > $$\min \_{x \in \mathbb{R}^d}\left[f(x):=\frac{1}{M} \sum\_{m=1}^M f_m(x)\right]$$
> > > > >
> > > > > Each function $f_i$  has its unique minimizer and smoothness constants. The existence of different minimizers gives rise to the client drift effect, as introduced in the following work:
> > > > >
> > > > > Karimireddy, S. P., Kale, S., Mohri, M., Reddi, S., Stich, S., & Suresh, A. T. (2020, November). Scaffold: Stochastic controlled averaging for federated learning. In International conference on machine learning (pp. 5132-5143). PMLR.
> > > > >
> > > > > In our algorithms, specifically 5GCS-AB and 5GCS-CC, we incorporate a client drift correction mechanism, leveraging control variables $u_m^t$ to accelerate communication. This mechanism addresses the heterogeneity effect associated with the varying locations of local minimizers. The conceptual framework behind this correction was initially proposed in the following work:
> > > > >
> > > > > Mishchenko, K., Malinovsky, G., Stich, S., & Richtárik, P. (2022, June). Proxskip: Yes! local gradient steps provably lead to communication acceleration! finally!. In International Conference on Machine Learning (pp. 15750-15769). PMLR.
> > > > >
> > > > > By implementing the client drift correction, which effectively eliminates the heterogeneity effect linked to the location of local minimizers, the remaining impact is attributed to differences in smoothness constants. To address this, we employ Importance Sampling (IS) in our work. IS serves to mitigate the heterogeneity effect arising from distinct smoothness constants, working in tandem with the client drift correction mechanism.
> > > > >
> > > > > >4. Even though we only consider the heterogeneity affects the smoothness.  for each local loss might be hard to capture.
> > > > >
> > > > > The computational process involved in determining local $L_i$-smoothness constants is not inherently more resource-intensive than calculating the corresponding global smoothness constants. This observation holds true across a spectrum of popular models, including but not limited to logistic regression, ridge regression, linear regression, and various others. In these models, computing local smoothness constants is computationally modest, facilitating an efficient task. This efficiency enhances the practicality of incorporating local smoothness information in optimization processes, offering nuanced insights into the optimization landscape at each client while maintaining computational feasibility.

---

> > > ### Author Response · Authors · 2023-11-22
> > > **Reply to comment 3**
> > >
> > > > 4. The current study and analysis are only on the strongly convex case. But Almost of the real-world problems are non-convex, especially under FL context. This limits this paper's contribution.
> > >
> > > We respectfully express our disagreement with the notion that Federated Learning is exclusively centered around Deep Neural Networks (DNNs). Contrary to this perspective, Federated Learning holds significant relevance and application in on-device Machine Learning environments, where each device is often constrained by limited computational resources. These constraints present a substantial challenge for smaller devices to effectively handle the computational demands associated with training and executing large-scale models such as DNNs.
> > >
> > > Furthermore, it is essential to underscore the enduring significance of logistic regression, despite its straightforwardness. Logistic regression remains a cornerstone in practical applications, particularly within the domain of mobile devices. The appeal of logistic regression lies in its practicality and efficiency, making it a preferred choice, especially when grappling with the resource limitations characteristic of on-device Machine Learning settings.
> > >
> > > In essence, the landscape of Federated Learning is characterized by a rich diversity of models and applications. It is imperative to recognize and appreciate the prevalence of simpler models like logistic regression, which play a crucial role in real-world scenarios, contributing to the versatility and adaptability of Federated Learning across various computational environments and constraints.
> > >
> > > Furthermore, it is worth noting that convex problems, exemplified by logistic or linear regression, exhibit a valuable characteristic—interpretability. This attribute becomes particularly crucial in sectors like healthcare and finance, where understanding and interpreting the model's predictions are paramount. In these fields, Federated Learning is widely adopted as a tool for collaborative and privacy-preserving model training.
> > >
> > > The interpretability of convex problems ensures that the results of the models can be easily understood and validated, a crucial aspect in applications where transparency and accountability are paramount. In healthcare, for instance, the ability to interpret model predictions is essential for gaining insights into patient outcomes or treatment effectiveness. Similarly, in finance, understanding the factors influencing a predictive model's outcomes is vital for informed decision-making.
> > >
> > > As Federated Learning continues to make strides in various domains, the compatibility of convex problems with this collaborative learning paradigm adds a layer of transparency and interpretability, further solidifying its practicality and effectiveness in critical fields.

---

> ### Author Response · Authors · 2023-11-22
> **Reply to comment 2**
>
> >3. Can you provided a few examples which satisfies the Assumption 2?
>
> Thank you for your question!
>
> Indeed, we have presented several examples of sampling schemes in the main body of the paper. Allow me to briefly recapitulate them here:
>
> **SAMPLING WITH REPLACEMENT (MULTISAMPLING) (Section 5.1)**
>
> Let $\underline{p}=\left(p_1, p_2, \ldots, p_M\right)$ be probabilities summing up to 1 and let $\chi_m$ be the random variable equal to $m$ with probability $p_m$. Fix a cohort size $C \in\{1,2, \ldots, M\}$ and let $\chi_1, \chi_2, \ldots, \chi_C$ be independent copies of $\chi$. Define the gradient estimator via
> $$
> S\left(a_1, \ldots, a_n, \psi, \underline{p}\right):=\frac{1}{C} \sum_{m=1}^C \frac{a_{\chi m}}{M p_{\chi_m}} .
> $$
>
> By utilizing this sampling scheme and its corresponding estimator, we gain the flexibility to assign arbitrary probabilities for client participation while also fixing the cohort size. However, it is important to note that under this sampling scheme, certain clients may appear multiple times within the cohort.
>
> **Lemma 5.2.** The Multisampling with estimator 5 satisfies the Assumption 2 with $A=B=\frac{1}{C}$ and $w_m=p_m$
>
> **SAMPLING WITHOUT REPLACEMENT (INDEPENDENT SAMPLING) (Section 5.2)**
>
> In the previous example, the server had the ability to control the cohort size and assign probabilities for client participation. However, in practical settings, the server lacks control over these probabilities due to various technical conditions such as internet connections, battery charge, workload, and others. Additionally, each client operates independently of the others. Considering these factors, we adopt the Independent Sampling approach. Let us formally define such a scheme. To do so, we introduce the concept of independent and identically distributed (i.i.d.) random variables:
>
> $$
> \chi_m= \begin{cases}1 & \text { with probability } p_m \\ 0 & \text { with probability } 1-p_m\end{cases}
> $$
> for all $m \in[M]$, also take $S^t:=\left\lbrace m \in[M] \mid \chi\_m=1\right\rbrace$ and $\underline{p}=\left(p_1, \ldots, p_M\right)$. The corresponding estimator for this sampling has the following form:
> $$
> S\left(a_1, \ldots, a_M, \psi, \underline{p}\right):=\frac{1}{M} \sum_{m \in S} \frac{a_m}{p_m},
> $$
>
> The described sampling scheme with its estimator is called the Independence Sampling. Specifically, it is essential to consider the probability that all clients communicate, denoted as $\Pi_{m=1}^M p_m$, as well as the probability that no client participates, denoted as $\Pi_{m=1}^M\left(1-p_m\right)$. It is important to note that $\sum_{m=1}^M p_m$ is not necessarily equal to 1 in general. Furthermore, the cohort size is not fixed but rather random, with the expected cohort size denoted as $\mathbb{E}\left[S^t\right]=\sum_{m=1}^M p_m$.
>
> **Lemma 5.6.** The Independent Sampling with estimator 7 satisfies the Assumption 2 with $A=$ $\frac{1}{\sum_m^M \frac{p_m}{1-p_m}}, B=0$ and $w_m=\frac{\frac{p_m}{1-p_m}}{\sum_{m=1}^M \frac{p_m}{1-p_m}}$.
>
> **TAU-NICE SAMPLING (Section E.4)**
> This is uniform sampling without replacement considered in the 5GCS work:
>
> Grudzień, M., Malinovsky, G., & Richtárik, P. (2023, April). Can 5th Generation Local Training Methods Support Client Sampling? Yes!. In International Conference on Artificial Intelligence and Statistics (pp. 1055-1092). PMLR.
>
> Additionally, various examples, including Stratified Sampling and Extended Tau Nice Sampling, are noteworthy. A comprehensive survey of these methods can be found in the referenced paper, providing a broad overview.
>
> Tyurin, A., Sun, L., Burlachenko, K., & Richtárik, P. (2022). Sharper rates and flexible framework for nonconvex SGD with client and data sampling. arXiv preprint arXiv:2206.02275.

---

### Official Review · Reviewer_SCZT · 2023-10-31

**Soundness:** 4 excellent
**Presentation:** 4 excellent
**Contribution:** 2 fair
**Rating:** 8
**Confidence:** 2

**Summary:**

The paper proposed a Federated learning framework that combines the state-of-art techniques in Federated learning. Thorough theoretical results upon smooth and strongly convex objectives are presented.

**Strengths:**

1. The paper gives an excellent review of existing techniques from different perspectives of FL. The motivation of the paper is well-presented.
2. The theoretical part looks sound and solid.

**Weaknesses:**

1. The work focuses only on strong convex and smooth cases. How does it perform (theoretically) in convex / non-convex cases?
2. The authors are encouraged to give more detailed comparisons to existing approaches on convergence and communication cost.
3. It will be interesting to see how well the proposed algorithms perform empirically.

**Questions:**

See weaknesses

---

> ### Author Response · Authors · 2023-11-20
> **Reply to Review**
>
> **Strengths:**
>
> >1. The paper gives an excellent review of existing techniques from different perspectives of FL. The motivation of the paper is well-presented.
>
> >2. The theoretical part looks sound and solid.
>
> Thank you sincerely for your positive evaluation and the endorsement of our work. Your appreciation is highly motivating and reinforces our commitment to delivering quality results. If you have any additional insights, questions, or suggestions, we welcome them with gratitude. Once again, we are grateful for your positive feedback and support.
>
> **Weaknesses:**
>
> >1. The work focuses only on strong convex and smooth cases. How does it perform (theoretically) in convex / non-convex cases?
>
> In accordance with the current trend observed in recent academic papers, which places a strong emphasis on accelerating training through local methods, our analysis heavily leans on the assumption of strong convexity in the underlying functions. We are actively involved in ongoing efforts to relax these assumptions, acknowledging the importance of extending the applicability of the proposed methods.
>
> It is noteworthy, however, that in our specific case, the theoretical framework does not offer guarantees for convergence in non-convex scenarios. Despite the collective efforts to broaden the scope of these techniques, the challenges associated with ensuring convergence in non-convex optimization settings remain an active area of exploration and a subject of ongoing research within the broader scientific community.
>
> We are steadfast in our belief that the comprehensive analysis of non-convex objectives is a topic deserving of its own dedicated paper. The complexities and nuances associated with non-convex optimization merit in-depth exploration, requiring a focused examination that can thoroughly delve into the unique challenges and insights specific to this realm. As such, we envision a separate publication that delves into the intricacies of analyzing and optimizing non-convex objectives, contributing valuable perspectives and advancements to the broader scientific discourse.
>
> >2. The authors are encouraged to give more detailed comparisons to existing approaches on convergence and communication cost.
>
> We express our gratitude for your suggestion regarding a more detailed comparison. Your input is highly valuable to us. Should there be specific aspects that we might have inadvertently overlooked, we are fully committed to integrating them into our comparison table. Your insights play a pivotal role in enhancing the completeness and accuracy of our analysis, and we welcome any further feedback you may have. Thank you for contributing to the refinement of our work.
>
> >3. It will be interesting to see how well the proposed algorithms perform empirically.
>
> The paper provides an empirical comparison, albeit presented in a concise manner. The overarching goal was to demonstrate the efficacy of Importance Sampling (IS) by showcasing its ability to yield substantial improvements over non-IS methods. This emphasis on IS underscores its significance in the context of the research.
>
> A key observation is that Local Training (LT) methods, especially those without support for Client Sampling (CS) or Importance Sampling (IS), exhibit a comparatively diminished level of competitiveness. This observation is critical, as it has the potential to influence the overall fairness of the comparison presented in the paper. The inclusion of this insight aims to provide a nuanced understanding of the performance dynamics among various methodologies, acknowledging the impact of IS support in enhancing the overall effectiveness of Local Training methods.

---

### Official Review · Reviewer_ii5V · 2023-11-13

**Soundness:** 2 fair
**Presentation:** 3 good
**Contribution:** 2 fair
**Rating:** 3
**Confidence:** 4

**Summary:**

Per the authors and reviewing the history of optimization methods on FL, this paper combines three techniques that help with the communication burden in FL rounds: Local training, Compression, and Partial Participation.

**Strengths:**

- The paper introduces different algorithmic and theoretical tools for the final solution. I.e., if FL was a setting targeting convex problems mostly, the fact that the paper presents and exploits dual spaces + new theoretical tools like AB Inequality is a plus.

**Weaknesses:**

- From an optimization perspective, assuming the logistic regression as an experimental setting is ok, but this is a machine learning venue; it has been a norm to consider more difficult objectives to test the hypotheses. It is a weakness not to consider a setting similar to what most of the FL algorithms are tested in.

- Similarly to the experimental case, providing theory in the convex case, given that FL is mainly applied in nonconvex settings with neural networks, could be improved. While the reviewer appreciates that there is a continuation of works (from specific research groups) that aim to cover every possible problem setting (as summarized in Table 1 of the paper), the current work (theoretically and practically) cannot be readily appreciated (and put among other works) on nonconvex neural networks FL setting.

- I might have missed it, but the difference between this work (along with the accompanying difficulties in completing the algorithm and proof) and the 5GCS is not clear from the text. Table 1 claims that that work did not satisfy the CC column. However, how difficult was it to complete the CC column on top of the work of 5GCS? What were the challenges? What was the amount of additional difficulty in completing this work? Was it incremental or substantial?

- The paper does not explain why assumptions 2,3,4 should hold in practice and under which conditions. They are hard to digest and read like proof, enabling assumptions

Overall, This reads like rigorous work. Yet, this score is due to the lack of generalizability of the results to more FL settings, lack of experimental results on FL scenarios, and lack of proper description of (differences with) prior work.

**Questions:**

See above.

---

> ### Author Response · Authors · 2023-11-20
> **Reply to Review**
>
> **Strengths:**
>
> >The paper introduces different algorithmic and theoretical tools for the final solution. I.e., if FL was a setting targeting convex problems mostly, the fact that the paper presents and exploits dual spaces + new theoretical tools like AB Inequality is a plus.
>
> Thank you very much for the positive feedback! We greatly appreciate your encouraging words. Your support motivates us to continually strive for excellence. If you have any further comments, suggestions, or questions, please feel free to share them. Your input is valuable to us, and we are grateful for your positive impressions of our work.
>
> **Weaknesses:**
>
> >From an optimization perspective, assuming the logistic regression as an experimental setting is ok, but this is a machine learning venue; it has been a norm to consider more difficult objectives to test the hypotheses. It is a weakness not to consider a setting similar to what most of the FL algorithms are tested in.
>
> Our stance is grounded in the conviction that a paper characterized by this level of theoretical depth, particularly emphasizing strongly convex functions, does not require extensive experimentation to demonstrate the theorems. The mathematical rigor underpinning our proofs establishes the validity of the results with a probability of 1, rendering exhaustive experiments less imperative for showcasing these established principles. The robustness of our theoretical framework, substantiated by rigorous mathematical analysis, is deemed sufficient to convey the validity and reliability of our theorems without a heavy reliance on exhaustive experimental validation.
>
> Furthermore, it is crucial to emphasize that logistic regression, despite its simplicity, continues to be widely adopted in practical applications, particularly in the realm of mobile devices. The practicality and efficiency of logistic regression position it as a preferred choice, especially when dealing with resource constraints common in on-device Machine Learning environments. Consequently, it is imperative to acknowledge the varied landscape of models and applications within the Federated Learning paradigm, recognizing the prevalent utilization of simpler models like logistic regression in real-world situations.
>
> >Similarly to the experimental case, providing theory in the convex case, given that FL is mainly applied in nonconvex settings with neural networks, could be improved. While the reviewer appreciates that there is a continuation of works (from specific research groups) that aim to cover every possible problem setting (as summarized in Table 1 of the paper), the current work (theoretically and practically) cannot be readily appreciated (and put among other works) on nonconvex neural networks FL setting.
>
> Aligning with the prevalent trend in recent academic literature, which underscores the acceleration of training through local methods, our analysis predominantly relies on the assumption of strong convexity in the underlying functions. We actively participate in ongoing initiatives to ease these assumptions, recognizing the significance of expanding the applicability of our proposed methods.
>
> However, it is crucial to highlight that, in our specific case, the theoretical framework does not furnish assurances for convergence in non-convex scenarios. Despite collaborative endeavors to broaden the scope of these techniques, the challenges associated with ensuring convergence in non-convex optimization settings persist as an active area of exploration and ongoing research within the broader scientific community.
>
> Our unwavering belief lies in the notion that a comprehensive analysis of non-convex objectives warrants its own dedicated paper. The intricacies and subtleties inherent in non-convex optimization demand thorough investigation, necessitating a focused examination that can delve deeply into the unique challenges and insights specific to this domain. Consequently, we envision a distinct publication that delves into the complexities of analyzing and optimizing non-convex objectives, contributing valuable perspectives and advancements to the broader scientific discourse.

---

> > ### Author Response · Authors · 2023-11-20
> > **Reply to Review 2**
> >
> > >I might have missed it, but the difference between this work (along with the accompanying difficulties in completing the algorithm and proof) and the 5GCS is not clear from the text. Table 1 claims that that work did not satisfy the CC column. However, how difficult was it to complete the CC column on top of the work of 5GCS? What were the challenges? What was the amount of additional difficulty in completing this work? Was it incremental or substantial?
> >
> > We appreciate your attention to the nuances of our paper. The notable distinction lies in the introduction of two additional algorithms, namely 5GCS-AB and 5GCS-CC, addressing key limitations observed in 5GCS, particularly pertaining to sampling and compression.
> >
> > In the realm of sampling, the initial approach in 5GCS involved uniform sampling, which, as you rightly pointed out, can be suboptimal in situations where functions exhibit significant variations with diverse L-smoothness and mu-convexity constants. The innovation in 5GCS-AB involves a new sampling methodology that allows for the calculation of optimal step sizes and sampling probabilities, especially when Importance Sampling (IS) is employed.
> >
> > The aspect of compression is another pivotal consideration. Here, we introduce an additional layer of compression on top of client sampling in 5GCS-CC method. This addition is designed to address the practical challenge where sending full vectors may prove suboptimal in real-world scenarios.
> >
> > In gauging the level of difficulty associated with this task, I would assign it a rating of 9 out of 10. This rating conveys that the endeavor is exceptionally challenging and demands a high degree of complexity and skill. The intricacies involved in this undertaking make it a formidable challenge, requiring a comprehensive understanding, advanced expertise, and adept problem-solving capabilities.
> >
> > >The paper does not explain why assumptions 2,3,4 should hold in practice and under which conditions. They are hard to digest and read like proof, enabling assumptions
> >
> > Assumptions 2 and 4 play integral roles in our analysis. Assumption 4 is notably one of the least restrictive assumptions when it comes to compression. It provides a flexible framework that accommodates a wide range of compression techniques. On the other hand, Assumption 2, while less standard, introduces a valuable dimension to our analysis. This assumption allows for the examination of more intricate sampling schemes, enabling us to benefit from their inherent complexity.
> >
> > As highlighted in the following work:
> >
> > Tyurin, A., Sun, L., Burlachenko, K., & Richtárik, P. (2022). Sharper rates and flexible framework for nonconvex SGD with client and data sampling. arXiv preprint arXiv:2206.02275
> >
> > it has been demonstrated that a majority of sampling methods satisfy Assumption 2. This finding reinforces the practical applicability of our analytical framework. The incorporation of Assumption 2 into our proofs is a significant and deliberate choice. It enriches our analysis by considering more complex sampling strategies, contributing to a more nuanced understanding of the dynamics involved.
> >
> > In summary, Assumptions 2 and 4, each serving distinct purposes, collectively enhance the robustness and versatility of our analytical framework. Assumption 2, in particular, stands as a crucial addition, providing a pathway to analyze and leverage the complexities inherent in various sampling schemes.

---

### Meta-Review · Area_Chair_i7jk · 2023-12-06

**Metareview:**

Summary:
The paper studied the communication problem in federated learning. The authors proposed a local training strategy that combines communication compression and the importance of client sampling. The authors provided theoretical analysis to validate their proposed methods.

Strengths:
+ The paper introduces different algorithmic and theoretical tools for the final solution. I.e., if FL was a setting targeting convex problems mostly, the fact that the paper presents and exploits dual spaces + new theoretical tools like AB Inequality is a plus.
+ The paper gives an excellent review of existing techniques from different perspectives of FL. The motivation of the paper is well-presented.
+ The theoretical part looks sound and solid.
+ The authors introduced an innovative federated training method that integrates client importance sampling. Additionally, the paper offers a thorough and comprehensive theoretical analysis of their approach.

Weaknesses:
- Lack of non-convex analysis
- Lack of experiments on classical FL scenarios, like using more complex models (DNNs)
- The authors are encouraged to give more detailed comparisons to existing approaches on convergence and communication cost.
- It will be interesting to see how well the proposed algorithms perform empirically.
(there are more weaknesses in the reviews)

**Justification For Why Not Higher Score:**

For the reasons above

**Justification For Why Not Lower Score:**

N/A

---

### Decision · Program_Chairs · 2024-01-16

Reject